# Liposomes with Caffeic Acid: Morphological and Structural Characterisation, Their Properties and Stability in Time

Ioana Lavinia Dejeu [1], Laura Grațiela Vicaș [2,*], Tunde Jurca [2], Alin Cristian Teușdea [3,*], Mariana Eugenia Mureșan [2], Luminița Fritea [2], Paula Svera [4], Gianina Adela Gabor [5], George Emanuiel Dejeu [2], Octavian Adrian Maghiar [2], Anca Salomea Bodea [1], Annamaria Pallag [2] and Eleonora Marian [2]

[1] Doctoral School of Biomedical Science, University of Oradea, 1 University Street, 410087 Oradea, Romania; ioana.dejeu@gmail.com (I.L.D.); salomea.bodea@gmail.com (A.S.B.)

[2] Faculty of Medicine and Pharmacy, University of Oradea, 1 Decembrie Street, 410073 Oradea, Romania; jurcatunde@yahoo.com (T.J.); marianamur2002@yahoo.com (M.E.M.); fritea_luminita@yahoo.com (L.F.); dejeu.george@gmail.com (G.E.D.); octimaghiar@gmail.com (O.A.M.); annamariapallag@gmail.com (A.P.); marianeleonora@yahoo.com (E.M.)

[3] Faculty of Environmental Protection, University of Oradea, 26th Gen. Magheru Avenue, 410087 Oradea, Romania

[4] INCEMC-National Institute for Research and Development in Electrochemistry and Condensed Matter—Timisoara, No.144 Dr. A. Paunescu Podeanu Street, 300569 Timisoara, Romania; paulasvera@gmail.com

[5] Faculty of Electrical Engineering and Information Technology, University of Oradea, 1 University Street, 410087 Oradea, Romania; gianina@rdsor.ro

* Correspondence: laura.vicas@gmail.com (L.G.V.); ateusdea@yahoo.co.uk (A.C.T.)

**Abstract:** Medical and pharmaceutical research has shown that liposomes are very efficient in transporting drugs to targets. In this study, we prepared six liposome formulas, three in which we entrapped caffeic acid (CA), and three with only phospholipids and without CA. Determination of entrapment efficiency (EE) showed that regardless of the phospholipids used, the percentage of CA entrapment was up to 76%. The characterization of the liposomes was performed using Dynamic Light Scattering (DLS), Atomic Force Microscopy (AFM), zeta potential and polydispersity and showed that about 75–99% of the liposomes had dimensions between $40 \pm 0.55$–$500 \pm 1.45$ nm. The size and zeta potential of liposomes were influenced by the type of phospholipid used to obtain them. CA release from liposomes was performed using a six-cell Franz diffusion system, and it was observed that the release of entrapped CA occurs gradually, the highest amount occurring in the first eight hours (over 80%), after which the release is much reduced. Additionally, the time stability of the obtained liposomes was analysed using univariate and multivariate statistical analysis. Therefore, liposomes offer great potential in CA entrapment.

**Keywords:** caffeic acid; liposomes; Atomic Force Microscopy; Dynamic Light Scattering; stability

## 1. Introduction

Caffeic acid (3,4-dihydroxycinnamic acid) is a biosynthetic derivative of phenylalanine and belongs to the class of phenolic acids, which are considered secondary plant metabolites produced naturally by almost all plants [1].

Caffeic acid has an important antioxidant activity [2], is hepatoprotective [3] and antibacterial [4], and is beneficial in cancer prevention and treatment [5], Alzheimer's disease [6], diabetes [7] and inflammatory diseases [8].

However, when polyphenols are extracted, they easily lose their bioactivity if not properly protected from certain factors such as oxygen, light and heat [9]. Encapsulation of active compounds in a protective matrix can be performed using technologies like coacervation, evaporation of the emulsion or through liposomes, thus ensuring their bioactivity does not change [10].

Liposomes are small artificial vesicles that have one or more layers, being able to incorporate a wide range of lipophilic and hydrophilic compounds [11]. Bioactive compounds can be incorporated into both the lipophilic and the hydrophilic compartment, depending on their affinity for water or the lipid membrane [12]. Another important advantage of using liposomes as an encapsulation system is observed in the gastrointestinal tract, where absorption is increased, thus increasing the bioavailability of the drug [13].

Currently, liposomes are an important part of medical and pharmaceutical research, being considered to be among the most effective carriers for the introduction of various medicinal substances into target cells [14]. Liposomes contain phospholipids. The most commonly used phospholipids are extracted from soy or egg yolk. Phospholipids are made up of a hydrophilic "head" containing three molecular components: choline, a phosphate group and glycerol, and two "tails", contained in the hydrophobic compartment, that form a long chain of essential fatty acids [15]. Depending on the method of preparation, the properties of liposomes, their shape, size, stability and drug loading efficiency can be influenced [16]. They can have different sizes, ranging from a few nanometers to micrometers: multilamellar vesicles (MLV, >250 nm), large unilamellar vesicles (LUV, 100–250 nm), and small unilamellar vesicles (SUV, 20–100 nm) [17,18]. The most commonly used liposome preparation method is the thin-film hydration method [19–21].

In general, the liposomes applied to medical use range between 50 and 450 nm [22]. Particulate systems technology offers excellent opportunities for the pharmaceutical industry, thus achieving an encapsulation and a controlled release of various substances, obtaining good bioavailability and stability especially in the case of sensitive substances.

There are a wide range of applications in the medical field due to nanoencapsulation technology and the benefits that liposomes can offer: increased efficacy, high biocompatibility, low immunogenicity, drug protection, prolonged half-life of the drug, and low toxicity [23]. The use of liposomes in topical applications also has the advantage that it can reduce local irritation [24].

In medicine, liposomal formulations are approved for intravenous [25], intramuscular [26,27] and oral administration [28] in anticancer, antifungal, and anti-inflammatory treatments. Their applicability has been extended to the food industry, where various antioxidants and some flavors have been encapsulated in liposomes.

The purpose of this paper was to obtain structural and morphological characterization and an evaluation of the properties and stability of liposomes with caffeic acid over time.

## 2. Materials and Methods

Caffeic acid (CA), cholesterol (CHL), sodium cholate (SC), and phosphatidylcholine from egg yolk (PC) were obtained from Sigma-Aldrich Chemie GmbH, Steinheim, Germany; 1,2-dipalmitoyl-sn-glycero-3-phosphocholine (DP-PC), 1,2-dimyristoyl-sn-glycero-3-phosphocholine (DM-PC) from Avanti Polar Lipids Co., Alabaster, Alabama, USA; methanol from Promochem, LGC Standards GmbH, Wesel, Germany, chloroform from Merck KGaA, Damstadt, Germany, and Triton X-100 from Sigma-Aldrich Co., St Louis, MO, USA and phosphate buffer from Farmachim 10 SRL, Ploiești, Romania. All substances used had adequate purity, attested by analysis bulletins issued by the manufacturer.

### 2.1. Preparation of Liposomes

Liposomes encapsulated with CA were prepared by the thin-film hydration method. The six liposome formulas that have been obtained are: DPPC, DMPC, CNA encapsulated with CA and, as control, eDPPC, eDMPC, eCNA—free of CA and using different phospholipids. The composition of the liposomes is described in Table 1.

Lipid dispersions were prepared by dissolving precise amounts of substances in 2 mL of chloroform, stirring until complete dissolution. The volatile fraction of the solvent was removed using a rotavapor (Heidolph Hei-VAP Precision—Platinum3, Heidolph Instruments Gmbh &Co. KG, Schwabach, Germany) under the following working conditions: temperature 40 °C, speed 80 rpm, and pressure 200 mBar until a uniform and thin lipid

film is obtained, which appears on balloon walls. The next step is the hydration of the lipid film which was performed with 2 mL phosphate buffer solution (pH = 7.4), followed by being vigorously hand-shaken. The dispersions were kept for 2 h at room temperature for stabilization and then were mechanically agitated using a centrifuge (Hettich Universal 320 R), the working conditions being: temperature 40 °C, speed 500 rpm, and time 20 min. After that, the samples were sonicated for 30 min at 25 °C in an ultrasonic bath (Elmasonic S 100H). All samples were stored in a refrigerator (4–8 °C) until analysis. There are many methods of formulating liposomes, one of them using a supercritical assisted technique [29,30].

**Table 1.** Quantities of substances used in the preparation of liposomes with CA.

| Type of Liposome | Mass of CA (mg) | The Amount of DP-PC (mg) | The Amount of DM-PC (mg) | The Amount of PC (mg) | The Amount of CHL (mg) | The Amount of SC (mg) |
|---|---|---|---|---|---|---|
| DPPC | 25 | 50 | - | 50 | 2.5 | - |
| eDPPC | - | 50 | - | 50 | 2.5 | - |
| DMPC | 25 | - | 50 | 50 | 2.5 | - |
| eDMPC | - | - | 50 | 50 | 2.5 | - |
| CNA | 25 | - | - | 80 | 2.5 | 20 |
| eCNA | - | - | - | 80 | 2.5 | 20 |

CA—caffeic acid, DPPC, DMPC, CAN—liposomes loaded with CA, eDPPC, eDMPC, eCNA—empty liposomes, CHL—cholesterol, SC—sodium cholate, PC—phosphatidylcholine, DP-PC—1,2-dipalmitoyl-sn-glycero-3-phosphocholine, DM-PC—1,2-dimyristoyl-sn-glycero-3-phosphocholine.

The liposomes obtained were characterized using physicochemical methods; determination of particle size, zeta potential and entrapment efficiency, Atomic Force Microscopy, and in vitro release studies of caffeic acid entrapped in liposomes were conducted, followed by a statistical analysis of the results obtained.

### 2.2. Determination of Entrapment Efficiency (EE%)

The EE of CA-encapsulated liposomes was determined using spectrophotometry. After centrifugation, the absorbance of the CA remaining in the supernatant was measured using an UV-VIS spectrophotometer, PG Instruments T70+. Then, the concentration was calculated from a calibration plot obtained for pure CA. EE was calculated using Equation (1) [9]:

$$EE\ (\%) = \frac{Tca - Ts}{Tca} * 100 \tag{1}$$

where $T_{ca}$ is the total CA used in the liposomes and $T_s$ is the total CA present in the supernatant.

### 2.3. Determination of Particle Size and Zeta Potential of CA-Loaded Liposomes

The zeta potential is an analytical measurement method for characterizing the surface of nanoparticles, and its measurements are based on the principles of scattered light [31,32]. The Dynamic Light Scattering method (DLS) was applied to determine the diameter, distribution and zeta potential of the formulated liposomes. Depending on the size of the liposomes, the distribution of vesicles in the body is influenced. If the size of the liposomes is large, the risk of them being taken up and degraded by the endoplasmic reticulum increases.

The composition of phospholipids and the pH of the environment show us whether the liposomes are positively, negatively or neutrally charged [33].

Dynamic Light Scattering ZEN 3690 and Zetasizer Nano ZS (Malvern Panalytical, Malvern, UK) were used to characterize the liposome samples by measurement size and zeta potential. The results were presented as an intensity-weighted and volume-weighted

distribution of particle diameters (d.nm). The volume distribution was chosen to compare three possible nano-levels that the liposomes can achieve: (1) very small vesicles; (2) large vesicles and (3) flocculated vesicles [34–39]. The corresponding diameter ranges were assigned as follows: (1) 30–150 nm, (2) 150–500 nm and (3) 500–6000 nm [38,40]. From the volume-weighted data was extracted the mean particle diameter (V.mean (d.nm)) and the proportion of the particles' nano-levels (prop.V_1, prop.V_2 and prop.V_3 (%)). In this way, the volume-weighted distribution performs a better description of the liposome's dominant nano-level and a facile comparison between the liposome's carrier molecules without and with CA encapsulation (i.e., Sample Factor levels).

### 2.4. Atomic Force Microscopy (AFM) Measurements

Morphological analyses were performed using a Scanning Probe Microscopy Platform (MultiView-2000 system, Nanonics Imaging Ltd., Jerusalem, Israel) using intermittent mode, in ambient conditions (20 °C). For this analysis, a scanner equipped with a silicone probe and coated with chrome was used, with a radius of 20 nm and a resonance frequency of 30–40 KHz. Prior to AFM analysis, all samples were sonicated for 60 min. From each sample, 0.2 mL was poured dropwise onto an AFM glass slide holder. Subsequently, the samples were allowed to dry at 25 °C for 60 min (with constant fan ventilation), followed by the drying process at a temperature of 20 °C. The same environmental conditions were maintained for 30 days.

### 2.5. In Vitro Release Studies of CA Entrapped in Liposomes

For the qualitative and quantitative evaluation of liposomes, we measured the release of CA from liposomes, using a system of six Franz diffusion cells (Microette-Hanson system, model 57-6AS9, Copley Scientific Ltd., Nottingham, UK), with a diffusion surface of 1.767 cm$^2$ and a volume of 6.5 mL for the receiver chamber. The receptor chamber in each diffusion cell was filled with phosphate buffer (pH 7.4) mixed with freshly prepared 30% ethanol. The synthetic membranes, made of polysulfone with a diameter of 25 mm and with a pore size of 0.45 μm—Tuffryn®, PALL Life Sciences HT-450, batch T72556, were hydrated by immersion in the receptor medium for 30 min before use, then mounted between the donor and acceptor compartment of the Franz diffusion cell. Approximately 0.500 g of each sample was brought into the diffusion cell capsule. The system was maintained at 32 ± 1 °C and the receptor medium was stirred continuously (600 rpm) using a magnetic stirrer to avoid the effects of the diffusion layer. 0.5 mL of the receptor solution was taken at various time intervals (30 min, 1, 2, 3, 4, 5, 6, 7, 8, 12, 24, and 48 h) and replaced with fresh receptor medium to maintain a constant volume (6.5 mL) during the test. The amount of CA released was determined using a UV-VIS spectrophotometric method, the reading being performed at 325 nm.

### 2.6. Statistical Analysis

The DOE (Design Of Experiment) considered two statistical factors and one interaction factor:

- Factor One: Sample, with six levels: DPPC, eDPPC, DMPC, eDMPC, CNA, eCNA. The "e" letter encodes the liposome vesicles without CA encapsulation (as from empty vesicles);
- Factor Two: Time, with three levels: d1 (day1), d15 (day15), d30 (day30);
- Factor Three: Sample*Time (interaction factor) with 18 levels: DPPC_d1, DPPC_d15, DPPC_d30, eDPPC_d1, eDPPC_d15, eDPPC_d30, DMPC_d1, DMPC_d15, DMPC_d30, eDMPC_d1, eDMPC_d15, eDMPC_d30, CNA_1, CNA_d15, CNA_d30, eCNA_d1, eCNA_d15, eCNA_d30.

In order to have quantitative comparisons of the liposome's nano-properties between samples with and without CA encapsulation, and furthermore between the different carrier molecule liposomes, at day1, day15 and day30 time stamps, univariate statistical analysis was carried out by two-way Analysis Of Variance (2w-ANOVA) ($p$ = 0.05).

A multivariate statistical sequence of several methods was applied to decide which liposome samples had simultaneously: an abundance of particles within the nano-levels (1) (i.e., prop.V_1 and prop.V_2 high levels combined with low values of Z-Ave, PdI, V.mean and I.mean) and (2) high values of roughness (i.e., high values of Sa ang Sq). The multivariate statistical sequence used consisted of: PCA (Principal Component Analysis), MANOVA ($p$ = 0.05) (Multivariate ANOVA) and AHC (Agglomerative Hierarchical Cluster analysis).

All sample parameter data were analysed in triplicate ($n$ = 3). The statistical calculus and graphing were done by Matlab software (MatWorks Inc., 1 Apple Hill Drive, Natick, MA, USA) with homemade subroutines including standardised statistical methods.

## 3. Results

### 3.1. Preparation of Liposomes with CA

CA is a hydrophilic substance, so to improve its skin penetrability, we synthesized three types of liposomes.

Vesicle formulations were obtained by combinations of phospholipids (PC, DPPC, DMPC) and cholesterol using a molar ratio of approximately 10:1 by a thin-film hydration technique. We choose these phospholipids because they are recommended for human use, being composed of chains of higher fatty acids with different lengths and different degrees of saturation, representing the lipid membrane units of liposomes [41]. Thus, colloidal particles with a membrane composed of phospholipids and cholesterol loaded with CA were obtained. Liposomes are inert carrier systems that have lipophilic components oriented towards the center of the vesicle and a double polar head oriented towards the inside and towards the outer surface. The cholesterol fraction is included to model the rigidity of lipid membranes and helps to improve the stability of liposomes.

Figure 1 shows the scheme of liposome synthesis with successive work steps.

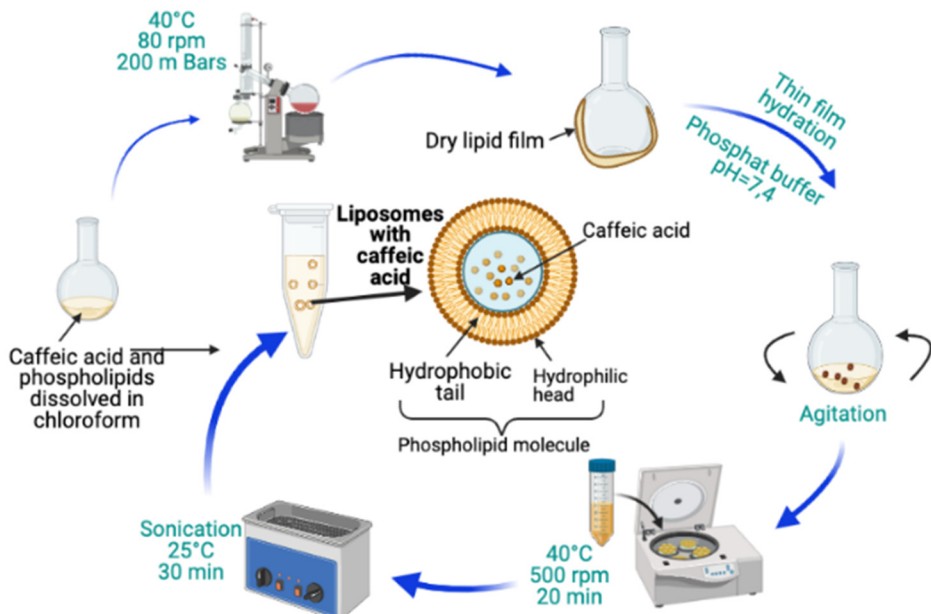

**Figure 1.** Preparation of liposomes with CA. CA—caffeic acid.

### 3.2. Entrapment Efficiency (EE%) of CA from Liposomes

UV-VIS spectrophotometry was used to determine the EE of the synthesized liposomes. The calibration curve for CA was used to interpret the results: $y$ = 0.0644$x$ + 0.0365, $R^2$ = 0.9997, where $y$—absorbance of the solution (u.a) at 325 nm, and $x$—concentration in CA (mmol/L).

The data obtained for the entrapment efficiency are shown in Table 2.

**Table 2.** EE of CA from the liposomes and zeta potential for liposomes.

| Liposome | DPPC | eDPPC | DMPC | eDMPC | CAN | eCNA |
|---|---|---|---|---|---|---|
| EE (%) | 75.22 ± 0.98 | - | 74.18 ± 1.01 | - | 75.93 ± 1.11 | - |
| Zeta Potential (mV) | −1.29 ± 0.03 | 0.56 ± 0.04 | −2.71 ± 0.03 | −2.48 ± 0.03 | −6.92 ± 0.02 | −10.90 ± 0.04 |

CA—caffeic acid, DPPC, DMPC, CNA—liposomes loaded with CA, eDPPC, eDMPC, eCNA—empty liposomes, and EE—entrapment efficiency.

### 3.3. Zeta Potential and the Size of the Liposomes with CA

Zeta-potential (mV) was determined by the instrument using a laser beam that passes through the center of the cell, into which the sample is inserted, detecting light scattered at a certain angle. The zeta potentials obtained for the six liposomes synthesized are shown in Table 2.

One of the research aims is the comparison between the liposomes' nano-levels without and with CA encapsulation. Figures 2–4 present intensity-weighted and volume-weighted nanoparticle distributions for each liposome's carrier molecules as Sample Factor levels, at the moment they were prepared (i.e., Time Factor Level Day 1).

**Intensity**          **Volume**

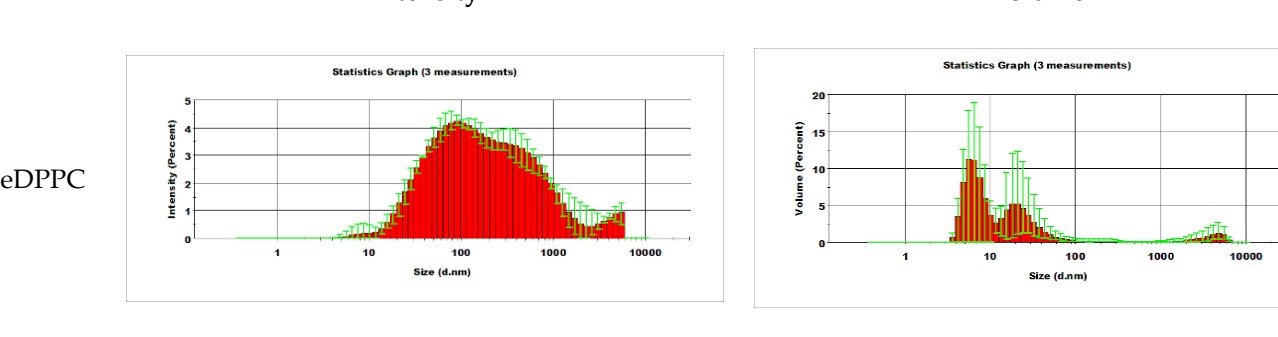

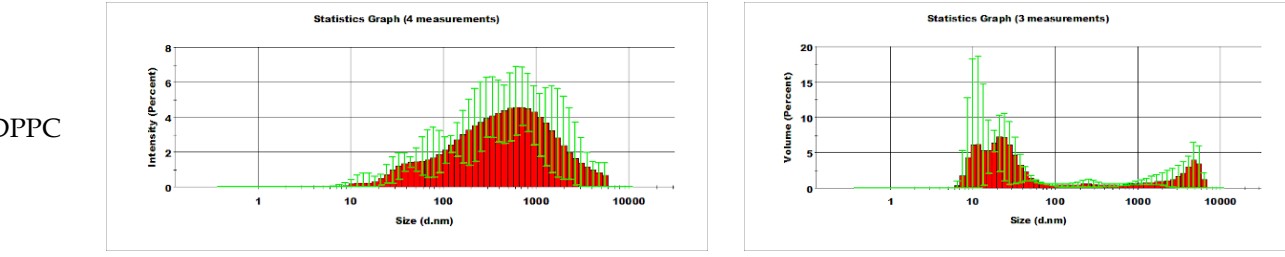

**Figure 2.** Intensity-weighted diameter distribution (**left column**) and volume-weighted radius distribution (**right column**) of the DPPC and eDPPC liposome variants. Green error bars denote the standard deviation of corresponding diameter bins. DPPC—liposomes loaded with CA, eDPPC—empty liposomes, and CA—caffeic acid.

**Intensity**　　　　　　　　　　　　　　　　　　　　　**Volume**

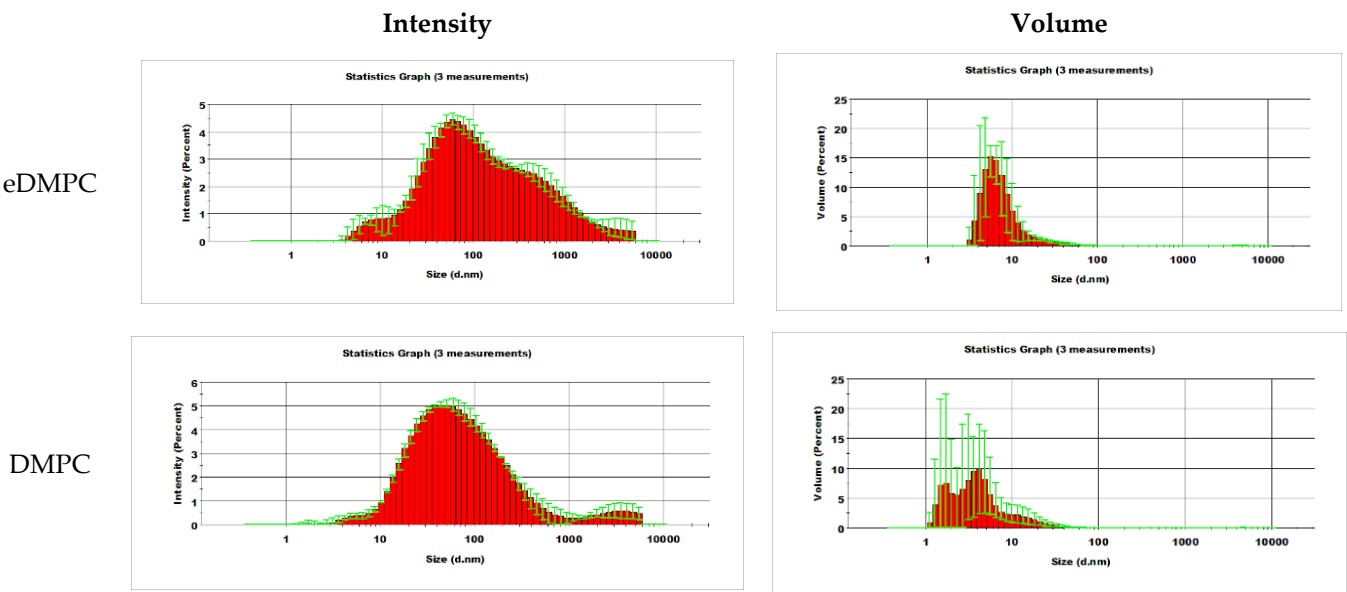

**Figure 3.** Intensity-weighted diameter distribution (**left column**) and volume-weighted radius distribution (**right column**) of the DMPC and eDMPC liposome variants. Green error bars denote the standard deviation of corresponding diameter bins. DMPC—liposomes loaded with CA, eDMPC—empty liposomes, and CA—caffeic acid.

**Intensity**　　　　　　　　　　　　　　　　　　　　　**Volume**

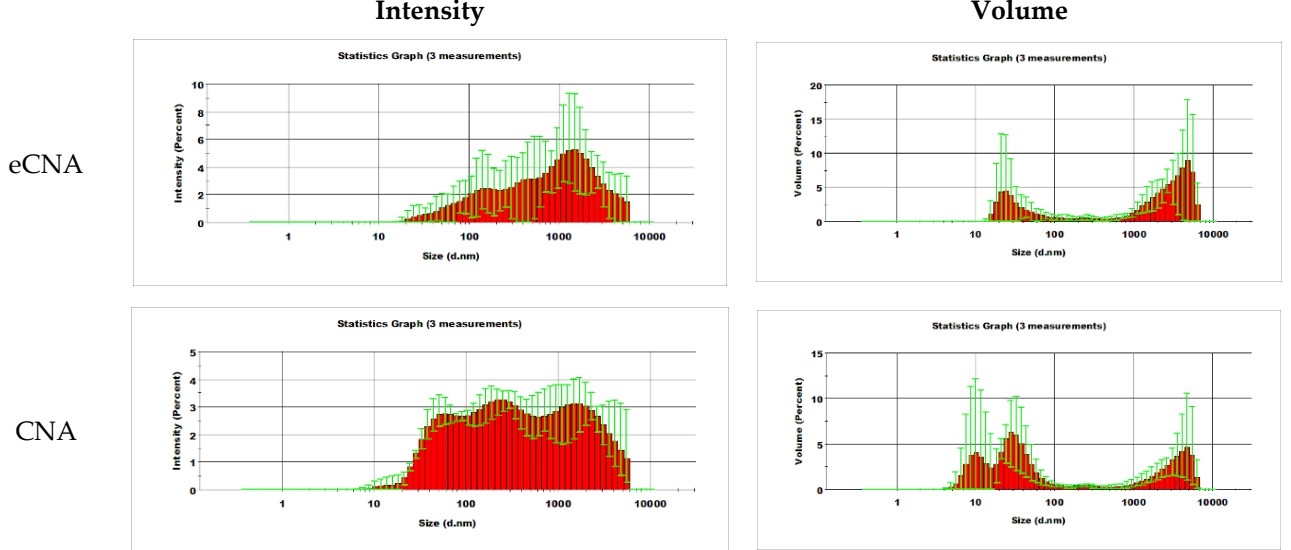

**Figure 4.** Intensity-weighted diameter distribution (**left column**) and volume-weighted radius distribution (**right column**) of the CNA and eCNA liposome variants. Green error bars denote the standard deviation of corresponding diameter bins. CA—caffeic acid, CNA—liposomes loaded with CA, and eCNA—empty liposomes.

### 3.4. Atomic Force Microscopy (AFM)

Description and characterization of liposomes by AFM were performed by the following measurements: average roughness (Sa), mean square root roughness (Sq), maximum peak height (Sp), naximum valley depth (Sv) and maximum peak to valley height (Sy). The values obtained for the six samples from the AFM images are represented in Tables 3–5. Measurements were performed at different time intervals in order to assess the stability of the liposomes. The intervals were: the first day (immediately after preparation), the second day, the 15th day and the 30th day.

**Table 3.** AFM values evolution for the sample DPPC and eDPPC.

| Sample | Day | Ironed Area (μm²) | Sa (μm) | Sq (μm) | Sp (μm) | Sv (μm) | Sy (μm) |
|--------|-----|-------------------|---------|---------|---------|---------|---------|
| DPPC | First day | 934.553 ± 0.122 | 0.1574 ± 0.0001 | 0.2135 ± 0.0007 | 0.7795 ± 0.0002 | −0.7733 ± 0.0003 | 1.5528 ± 0.0009 |
| | Second day | 934.925 ± 0.117 | 0.1590 ± 0.0001 | 0.2929 ± 0.0008 | 0.7949 ± 0.0005 | −0.7711 ± 0.0002 | 1.5660 ± 0.0008 |
| | 15th day | 946.628 ± 0.129 | 0.2146 ± 0.0001 | 0.2898 ± 0.0008 | 0.8504 ± 0.0002 | −1.3012 ± 0.0003 | 2.1517 ± 0.0011 |
| | 30th day | 940.870 ± 0.130 | 0.2059 ± 0.0002 | 0.0526 ± 0.0003 | 0.8742 ± 0.0007 | −1.3684 ± 0.0008 | 2.2426 ± 0.0010 |
| eDPPC | First day | 901.388 ± 0.103 | 0.0422 ± 0.0000 | 0.0571 ± 0.0003 | 0.1787 ± 0.0004 | −0.1776 ± 0.0006 | 0.3564 ± 0.0007 |
| | Second day | 901.388 ± 0.100 | 0.0422 ± 0.0001 | 0.0575 ± 0.0004 | 0.1787 ± 0.0003 | −0.1776 ± 0.0004 | 0.3564 ± 0.0009 |
| | 15th day | 902.115 ± 0.107 | 0.0460 ± 0.0000 | 0.2135 ± 0.0007 | 0.1912 ± 0.0006 | −0.2176 ± 0.0008 | 0.4089 ± 0.0009 |
| | 30th day | 902.142 ± 0.114 | 0.0460 ± 0.0000 | 0.2929 ± 0.0008 | 0.1812 ± 0.0005 | −0.2289 ± 0.0009 | 0.4101 ± 0.0010 |

CA—caffeic acid, DPPC—liposomes loaded with CA, and eDPPC—empty liposomes.

**Table 4.** AFM values evolution for the sample DMPC and eDMPC.

| Sample | Day | Ironed Area (μm²) | Sa (μm) | Sq (μm) | Sp (μm) | Sv (μm) | Sy (μm) |
|--------|-----|-------------------|---------|---------|---------|---------|---------|
| DMPC | First day | 916.767 ± 0.119 | 0.2026 ± 0.0001 | 0.2668 ± 0.0012 | 1.1410 ± 0.0012 | −0.7162 ± 0.0009 | 1.8572 ± 0.0011 |
| | Second day | 917.295 ± 0.123 | 0.1945 ± 0.0001 | 0.2537 ± 0.0010 | 1.1828 ± 0.0010 | −0.7599 ± 0.0007 | 1.9427 ± 0.0011 |
| | 15th day | 917.423 ± 0.120 | 0.1901 ± 0.0002 | 0.2483 ± 0.0010 | 1.2005 ± 0.0011 | −0.7561 ± 0.0008 | 1.9566 ± 0.0012 |
| | 30th day | 915.540 ± 0.121 | 0.1922 ± 0.0001 | 0.2409 ± 0.0009 | 0.9925 ± 0.0008 | −0.6398 ± 0.0007 | 1.6322 ± 0.0012 |
| eDMPC | First day | 904.453 ± 0.110 | 0.1357 ± 0.0001 | 0.1661 ± 0.0008 | 0.4717 ± 0.0007 | −0.4425 ± 0.0005 | 0.9143 ± 0.0011 |
| | Second day | 904.809 ± 0.114 | 0.1369 ± 0.0001 | 0.1690 ± 0.0011 | 0.5228 ± 0.0005 | −0.4330 ± 0.0004 | 0.9558 ± 0.0010 |
| | 15th day | 905.976 ± 0.115 | 0.1374 ± 0.0001 | 0.1708 ± 0.0010 | 0.9659 ± 0.0011 | −0.4411 ± 0.0006 | 0.9659 ± 0.0012 |
| | 30th day | 906.941 ± 0.118 | 0.1388 ± 0.0001 | 0.1713 ± 0.0008 | 0.5176 ± 0.0008 | −0.4332 ± 0.0005 | 0.9509 ± 0.0013 |

CA—caffeic acid, DMPC—liposomes loaded with CA, and eDMPC—empty liposomes.

**Table 5.** AFM values evolution for the sample CNA and eCNA.

| Sample | Day | Ironed Area (μm²) | Sa (μm) | Sq (μm) | Sp (μm) | Sv (μm) | Sy (μm) |
|--------|-----|-------------------|---------|---------|---------|---------|---------|
| CNA | First day | 902.074 ± 0.106 | 0.0689 ± 0.0001 | 0.0862 ± 0.0003 | 0.3379 ± 0.0003 | −0.2472 ± 0.0002 | 0.5852 ± 0.0005 |
| | Second day | 902.585 ± 0.109 | 0.0770 ± 0.0004 | 0.1000 ± 0.0002 | 0.3084 ± 0.0003 | −0.4049 ± 0.0005 | 0.7133 ± 0.0007 |
| | 15th day | 902.452 ± 0.101 | 0.0785 ± 0.0001 | 0.1003 ± 0.0004 | 0.3032 ± 0.0006 | −0.3223 ± 0.0006 | 0.6255 ± 0.0007 |
| | 30th day | 902.556 ± 0.107 | 0.0798 ± 0.0001 | 0.1015 ± 0.0005 | 0.3179 ± 0.0003 | −0.3152 ± 0.0004 | 0.6331 ± 0.0008 |
| eCNA | First day | 904.494 ± 0.104 | 0.0593 ± 0.0000 | 0.0744 ± 0.0004 | 0.2176 ± 0.0004 | −0.2594 ± 0.0004 | 0.4770 ± 0.0005 |
| | Second day | 903.352 ± 0.104 | 0.0623 ± 0.0004 | 0.0793 ± 0.0003 | 0.2524 ± 0.0003 | −0.3342 ± 0.0003 | 0.5867 ± 0.0005 |
| | 15th day | 902.616 ± 0.105 | 0.0645 ± 0.0000 | 0.0814 ± 0.0003 | 0.2523 ± 0.0007 | −0.3766 ± 0.0007 | 0.6289 ± 0.0009 |
| | 30th day | 903.604 ± 0.104 | 0.0668 m ± 0.0001 | 0.0866 ± 0.0003 | 0.3511 ± 0.0006 | −0.3127 ± 0.0006 | 0.6638 ± 0.0010 |

CA—caffeic acid, CNA—liposomes loaded with CA, and eCNA—empty liposomes.

To observe the time stability of the synthesized liposomes, 2D and 3D determinations were performed at time intervals: day 1, day 2, day 15, and day 30 (Figure 5).

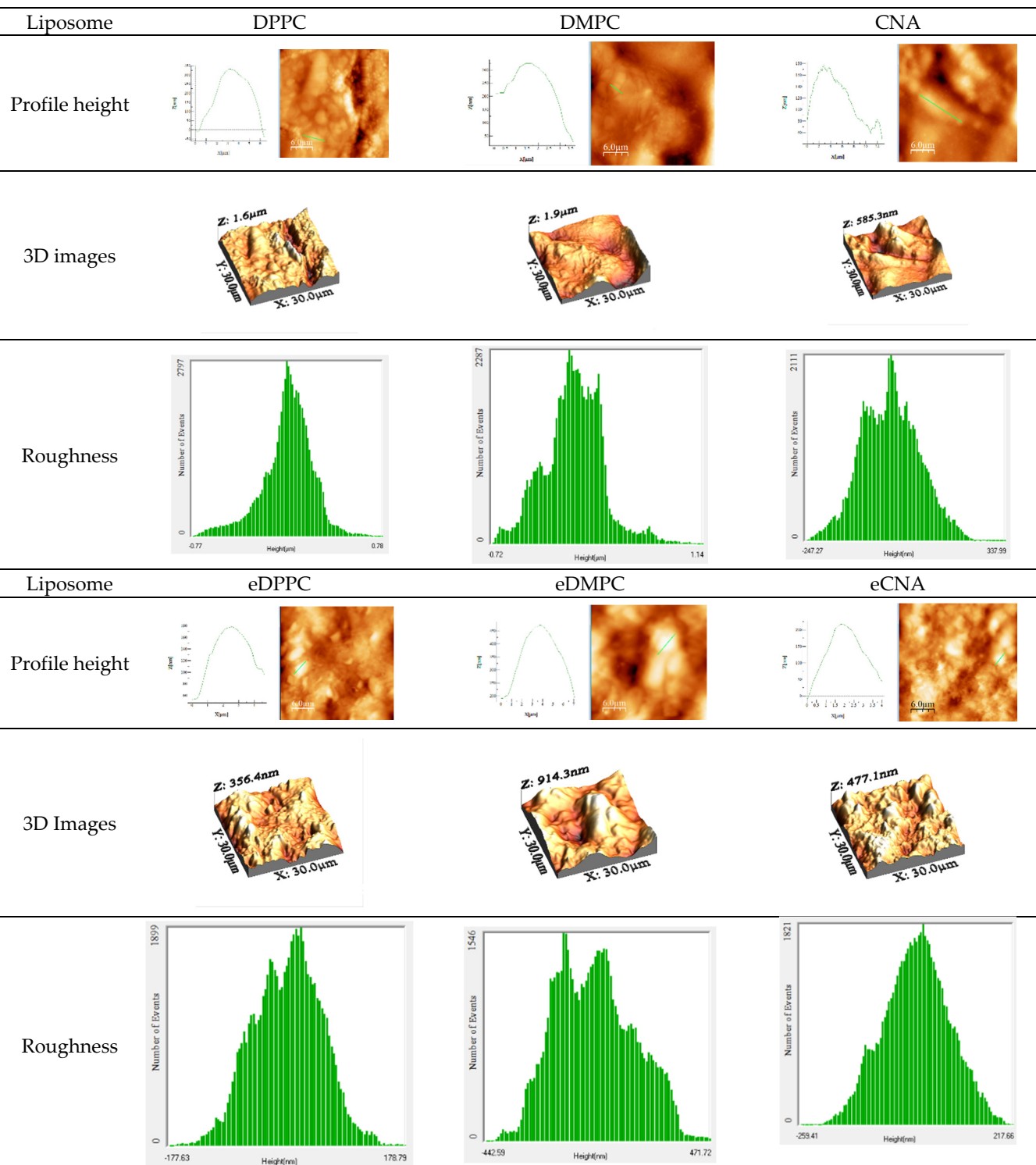

**Figure 5.** AFM images obtained for liposomes DPPC, DMPC, CNA, eDPPC, eDMPC, eCNA (day 1); CA—caffeic acid, DPPC, DMPC, CNA—liposomes loaded with CA, and eDPPC, eDMPC, eCNA—empty liposomes.

### 3.5. In Vitro Release Studies of CA Entrapped in Liposomes

To determine the CA entrapped in the synthesized liposomes, samples were taken from Franz cells at different times: 0.5, 1, 2, 3, 4, 5, 6, 7, 8, 12, 24, and 48 h. For determining the CA released from liposomes, the calibration curve of the CA from EE was used. It can be seen that the release of entrapped CA takes place gradually, with the largest amount

taking place in the first 8 h, after which the trajectory of the curves becomes almost parallel to the abscissa, the release being much reduced. The graphical representation of CA release from liposomes is illustrated in Figure 6.

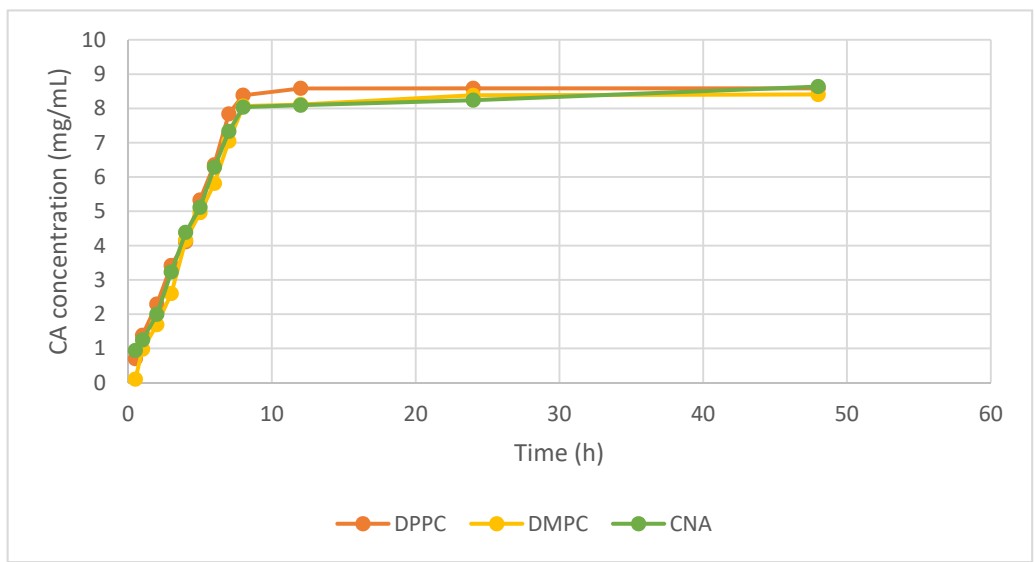

**Figure 6.** The release of CA from synthesized liposomes: red—DPPC, yellow—DMPC, green—CNA; CA—caffeic acid, and DPPC, DMPC, CNA—liposomes loaded with CA.

The release of CA from synthesized liposomes, related to the amount of CA entrapped in each liposome, over 48 h is shown in Table 6. For comparison, free CA was also introduced into the diffusion cell.

**Table 6.** Percentage of CA released from liposomes synthesized as a function of time and amount entrapped in each liposome.

| Mass of CA Released (%) from Liposomes | Time (h) | | | | | | | | | | | |
|---|---|---|---|---|---|---|---|---|---|---|---|---|
| | 0.5 | 1 | 2 | 3 | 4 | 5 | 6 | 7 | 8 | 12 | 24 | 48 |
| DPPC | 7.313 | 14.435 | 23.932 | 35.709 | 42.926 | 55.652 | 66.384 | 81.864 | 87.562 | 89.652 | 89.653 | 89.654 |
| DMPC | 1.039 | 10.123 | 18.691 | 27.057 | 43.234 | 51.559 | 60.452 | 73.318 | 83.914 | 84.413 | 87.211 | 87.452 |
| CAN | 9.717 | 12.924 | 20.660 | 33.490 | 45.471 | 53.018 | 65.188 | 76.037 | 83.377 | 83.956 | 85.483 | 85.624 |
| CA | 10.334 | 24.242 | 54.875 | 71.982 | 92.534 | 92.884 | 92.899 | 93.002 | 93.127 | 93.251 | 93.273 | 93.296 |

CA—caffeic acid, DPPC, DMPC, CNA—liposomes loaded with CA, and eDPPC, eDMPC, eCNA—empty liposomes.

### 3.6. Statistical Analysis Results

#### 3.6.1. Univariate Analysis

The univariate analysis (2w-ANOVA, $p = 0.05$) results are presented in Tables 7 and 8 and Figures 7–10, for each analysed statistical factor. Table 8 contains the mean sample values with standard deviations, along with the statistical significances from Duncan test ($p = 0.05$) post-hoc pairwise comparisons of mean sample values.

**Table 7.** AFM and DLS analysis results of liposomes before (eDPPC, eDMPC and eCNA) and after caffeic acid encapsulation (DPPC, DMPC and CNA). The results belong to the Sample factor levels (means ± SD, *n* = 3).

| Factor: Sample | Sa | Sq | PdI | Z-Ave (d.nm) | V.Mean (d.nm) | prop.V_1 (%) | prop.V_2 (%) | prop.V_3 (%) | I.Mean (d.nm) |
|---|---|---|---|---|---|---|---|---|---|
| DPPC | 0.1926 [b] ± 0.0267 | 0.2654 [a] ± 0.0390 | 0.68 [b] ± 0.18 | 279 [b] ± 78 | 910 [b] ± 381 | 65.06 [c] ± 20.50 | 3.14 [ab] ± 3.09 | 26.83 [a] ± 24.26 | 828 [a] ± 196 |
| eDPPC | 0.0448 [f] ± 0.0019 | 0.0558 [f] ± 0.0024 | 0.76 [b] ± 0.13 | 125 [d] ± 46 | 160 [c] ± 147 | 93.57 [a] ± 4.36 | 1.12 [bc] ± 0.76 | 4.09 [b] ± 4.19 | 467 [c] ± 56 |
| DMPC | 0.1950 [a] ± 0.0058 | 0.2520 [b] ± 0.0116 | 0.57 [c] ± 0.13 | 49 [f] ± 7 | 28 [c] ± 23 | 99.38 [a] ± 0.58 | 0.07 [c] ± 0.10 | 0.43 [b] ± 0.55 | 281 [d] ± 84 |
| eDMPC | 0.1373 [c] ± 0.0013 | 0.1694 [c] ± 0.0026 | 0.86 [a] ± 0.05 | 99 [e] ± 34 | 104 [c] ± 107 | 86.73 [ab] ± 18.02 | 0.64 [c] ± 1.12 | 1.14 [b] ± 1.98 | 478 [c] ± 170 |
| CNA | 0.0757 [d] ± 0.0052 | 0.0960 [d] ± 0.0074 | 0.91 [a] ± 0.12 | 195 [c] ± 32 | 1418 [a] ± 844 | 50.12 [d] ± 18.25 | 5.25 [a] ± 3.26 | 34.30 [a] ± 18.30 | 861 [a] ± 145 |
| eCNA | 0.0635 [e] ± 0.0033 | 0.0808 [e] ± 0.0053 | 0.75 [b] ± 0.14 | 306 [a] ± 60 | 403 [c] ± 320 | 74.93 [bc] ± 23.97 | 1.17 [bc] ± 2.82 | 23.43 [a] ± 23.51 | 707 [b] ± 192 |

CA—caffeic acid, DPPC, DMPC, CNA—liposomes loaded with CA, and eDPPC, eDMPC, eCNA—empty liposomes. Note: Different letters that follow the means prescribe statistically significant means. Results were calculated with *post-hoc* Duncan (*p* = 0.05) multiple comparisons test, within the two-way ANOVA test (*p* = 0.05).

**Table 8.** AFM and DLS analysis results of liposomes before (eDPPC, eDMPC and eCNA) and after CA encapsulation (DPPC, DMPC and CNA). The results belong to the Sample*Time interaction factor levels (means ± SD, *n* = 3).

| Factor: Sample*Time | Sa | Sq | PdI | Z-Ave (d.nm) | V.Mean (d.nm) | prop.V_1 (%) | prop.V_2 (%) | prop.V_3 (%) | I.Mean (d.nm) |
|---|---|---|---|---|---|---|---|---|---|
| DPPC_d1 | 0.1574 [f] ± 0.0001 | 0.2135 [f] ± 0.0007 | 0.8660 [abc] ± 0.2218 | 181.37 [de] ± 10.71 | 883.96 [bcd] ± 706.77 | 70.7863 [bcd] ± 20.9769 | 4.1567 [ab] ± 4.2525 | 16.5307 [bcd] ± 19.5549 | 699.43 [bcd] ± 194.59 |
| DPPC_d15 | 0.2146 [a] ± 0.0001 | 0.2929 [a] ± 0.0008 | 0.6240 [f] ± 0.0401 | 307.03 [b] ± 35.03 | 777.40 [bcde] ± 108.38 | 54.5470 [de] ± 27.7774 | 2.1597 [b] ± 2.3288 | 36.9633 [ab] ± 37.8118 | 738.63 [bc] ± 62.85 |
| DPPC_d30 | 0.2059 [b] ± 0.0002 | 0.2898 [b] ± 0.0008 | 0.5622 [fg] ± 0.1102 | 349.23 [a] ± 13.05 | 1069.54 [abc] ± 69.59 | 69.8567 [bcd] ± 14.8801 | 3.1300 [ab] ± 3.4198 | 27.0133 [abcd] ± 15.1663 | 1045.94 [a] ± 68.51 |
| eDPPC _d1 | 0.0422 [q] ± 0.0000 | 0.0526 [o] ± 0.0003 | 0.8073 [bcd] ± 0.0534 | 90.3167 [hi] ± 2.4871 | 288.42 [cde] ± 180.47 | 91.4567 [ab] ± 5.7847 | 1.2955 [b] ± 0.8758 | 4.4051 [d] ± 6.1183 | 468.87 [efg] ± 29.58 |
| eDPPC _d15 | 0.0460 [p] ± 0.0000 | 0.0571 [n] ± 0.0003 | 0.8783 [abc] ± 0.0455 | 111.13 [gh] ± 3.14 | 112.79 [de] ± 129.93 | 96.3663 [a] ± 2.7551 | 0.5560 [b] ± 0.8389 | 1.0987 [d] ± 1.9029 | 513.73 [def] ± 64.33 |
| eDPPC _d30 | 0.0460 [p] ± 0.0000 | 0.0575 [n] ± 0.0004 | 0.5943 [fg] ± 0.0247 | 174.47 [ef] ± 53.59 | 81.0301 [de] ± 10.7688 | 92.8931 [ab] ± 4.0021 | 1.5294 [b] ± 0.3094 | 6.7914 [d] ± 2.2227 | 420.22 [efgh] ± 31.62 |
| DMPC_d1 | 0.2026 [c] ± 0.0001 | 0.2668 [c] ± 0.0012 | 0.5527 [fg] ± 0.1328 | 44.1200 [j] ± 1.6975 | 13.2457 [e] ± 8.4025 | 99.80 [a] ± 0.13 | 0.0120 [b] ± 0.0208 | 0.1564 [d] ± 0.1792 | 264.03 [gh] ± 63.38 |
| DMPC_d15 | 0.1901 [e] ± 0.0002 | 0.2483 [d] ± 0.0010 | 0.4557 [g] ± 0.0130 | 45.4200 [j] ± 1.1677 | 26.0023 [e] ± 33.8572 | 99.49 [a] ± 0.74 | 0.1019 [b] ± 0.1613 | 0.3583 [d] ± 0.6207 | 212.40 [h] ± 62.46 |
| DMPC_d30 | 0.1922 [d] ± 0.0001 | 0.2409 [e] ± 0.0009 | 0.7167 [cdef] ± 0.0455 | 59.8533 [ij] ± 3.4546 | 46.5200 [e] ± 13.4920 | 98.8633 [a] ± 0.2838 | 0.0998 [b] ± 0.0878 | 0.8037 [d] ± 0.6962 | 369.00 [fgh] ± 40.88 |
| eDMPC_d1 | 0.1357 [i] ± 0.0001 | 0.1661 [h] ± 0.0008 | 0.8527 [abc] ± 0.0509 | 58.0267 [ij] ± 1.4662 | 23.7200 [e] ± 3.5127 | 99.38 [a] ± 0.08 | 0.1176 [b] ± 0.0486 | 0.3436 [d] ± 0.3035 | 360.90 [fgh] ± 58.99 |
| eDMPC_d15 | 0.1374 [h] ± 0.0001 | 0.1708 [g] ± 0.0010 | 0.9033 [ab] ± 0.0483 | 103.49 [gh] ± 3.46 | 70.4377 [de] ± 74.8932 | 97.4220 [a] ± 3.1617 | 0.0635 [b] ± 0.0985 | 2.4840 [d] ± 3.2319 | 631.43 [cde] ± 27.05 |
| eDMPC_d30 | 0.1388 [g] ± 0.0001 | 0.1713 [g] ± 0.0008 | 0.8393 [abcd] ± 0.0415 | 137.30 [fg] ± 8.52 | 220.55 [de] ± 93.63 | 63.4008 [cde] ± 7.8501 | 1.7484 [b] ± 1.5217 | 0.6110 [d] ± 1.0583 | 444.60 [efg] ± 232.91 |
| CNA_d1 | 0.0689 [l] ± 0.0001 | 0.0862 [k] ± 0.0003 | 0.9587 [ab] ± 0.0335 | 152.97 [ef] ± 11.03 | 1177.20 [ab] ± 902.34 | 63.9533 [cde] ± 20.8190 | 2.5593 [ab] ± 1.1769 | 33.4900 [abc] ± 19.7914 | 915.00 [ab] ± 183.50 |
| CNA_d15 | 0.0785 [k] ± 0.0001 | 0.1003 [j] ± 0.0004 | 0.9757 [a] ± 0.0421 | 212.93 [cd] ± 7.38 | 1341.00 [ab] ± 245.78 | 41.7467 [e] ± 2.0213 | 6.6177 [a] ± 2.8428 | 51.6367 [a] ± 2.9538 | 913.53 [ab] ± 109.69 |
| CNA_d30 | 0.0798 [j] ± 0.0001 | 0.1015 [i] ± 0.0005 | 0.7970 [bcde] ± 0.1728 | 220.63 [c] ± 1.22 | 1735.90 [a] ± 1316.12 | 44.6633 [e] ± 21.4107 | 6.6017 [a] ± 4.0827 | 17.8000 [bcd] ± 8.9643 | 756.87 [bc] ± 119.51 |
| eCNA_d1 | 0.0593 [o] ± 0.0000 | 0.0744 [m] ± 0.0004 | 0.6407 [ef] ± 0.0429 | 338.27 [ab] ± 21.41 | 286.51 [cde] ± 217.69 | 81.0097 [abc] ± 12.8173 | 0.6897 [b] ± 1.1945 | 12.1417 [bcd] ± 16.5246 | 795.87 [bc] ± 254.21 |
| eCNA_d15 | 0.0645 [n] ± 0.0000 | 0.0814 [l] ± 0.0003 | 0.6837 [def] ± 0.0705 | 342.97 [ab] ± 59.08 | 756.80 [bcde] ± 267.20 | 48.4723 [de] ± 20.1293 | 2.8273 [ab] ± 4.8971 | 48.7000 [a] ± 20.8312 | 808.07 [bc] ± 53.38 |
| eCNA_d30 | 0.0668 [m] ± 0.0001 | 0.0866 [k] ± 0.0003 | 0.9433 [ab] ± 0.0496 | 237.57 [c] ± 5.99 | 167.11 [de] ± 20.82 | 95.3119 [a] ± 1.3438 | 0.0000 [b] ± 0.0000 | 9.4719 [cd] ± 7.9926 | 518.40 [def] ± 30.14 |

CA—caffeic acid, DPPC, DMPC, CNA—liposomes loaded with CA, and eDPPC, eDMPC, eCNA—empty liposomes. Note: Different letters that follow the means prescribe statistically significant means. Results were calculated with *post-hoc* Duncan (*p* = 0.05) multiple comparisons test, within the two-way ANOVA test (*p* = 0.05).

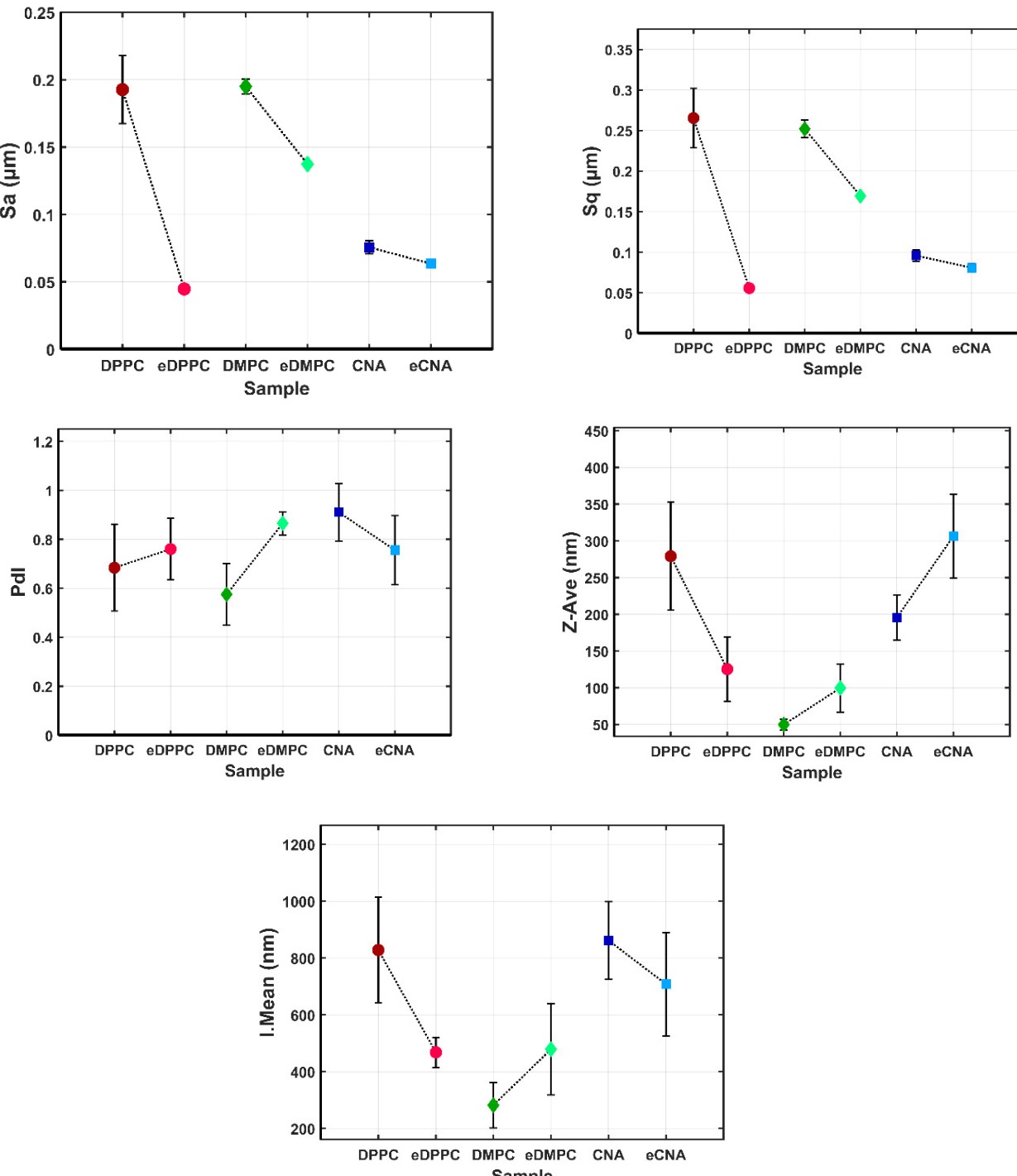

**Figure 7.** Interval plots of the liposome's properties for Sample factor levels; CA—caffeic acid, DPPC, DMPC, CNA—liposomes loaded with CA, and eDPPC, eDMPC, eCNA—empty liposomes.

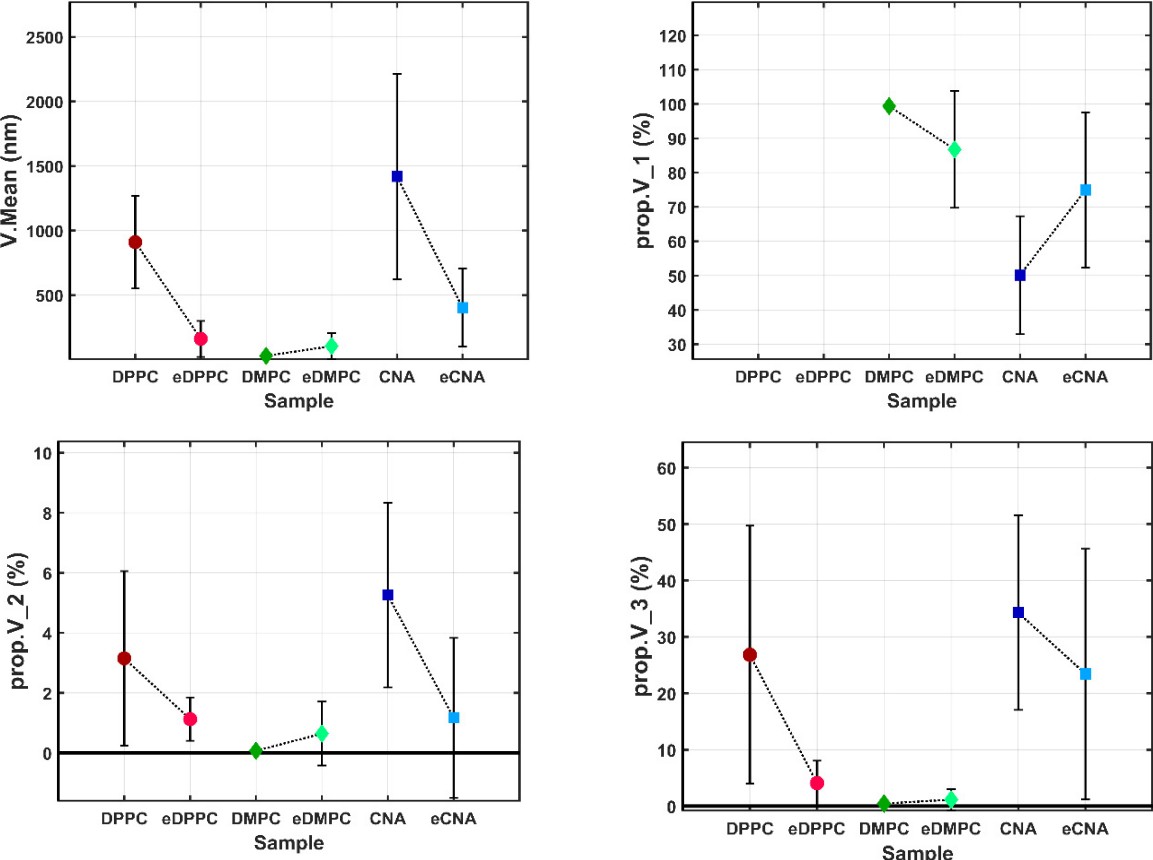

**Figure 8.** Interval plots of the liposome's properties derived from DLS analysis's volume-weighted distribution. Results are displayed for the Sample factor levels; CA—caffeic acid, DPPC, DMPC, CNA—liposomes loaded with CA, and eDPPC, eDMPC, eCNA—empty liposomes.

For qualitative comparisons the I.mean and V.mean particles diameter results are presented in Tables 7–9 and Figures 7 and 8. These parameters describe the volume-weighted particle size changes when the CA is encapsulated.

**Table 9.** PCA analysis data summary. Values in bold face denotes the retained PC's that ensures the cumulative explained variance greater than 95%.

| PC | Eigenvalue | % Variance | Cumulative % Variance |
|---|---|---|---|
| **1** | **3.364** | **48.062** | **48.062** |
| **2** | **2.167** | **30.961** | **79.023** |
| **3** | **0.895** | **12.789** | **91.812** |
| **4** | **0.394** | **5.624** | **97.436** |
| 5 | 0.120 | 1.716 | 99.152 |
| 6 | 0.058 | 0.831 | 99.982 |
| 7 | 0.001 | 0.018 | 100.000 |

The Sample*Time factor results averages all the values for each sample (Table 8 and Figures 9 and 10).

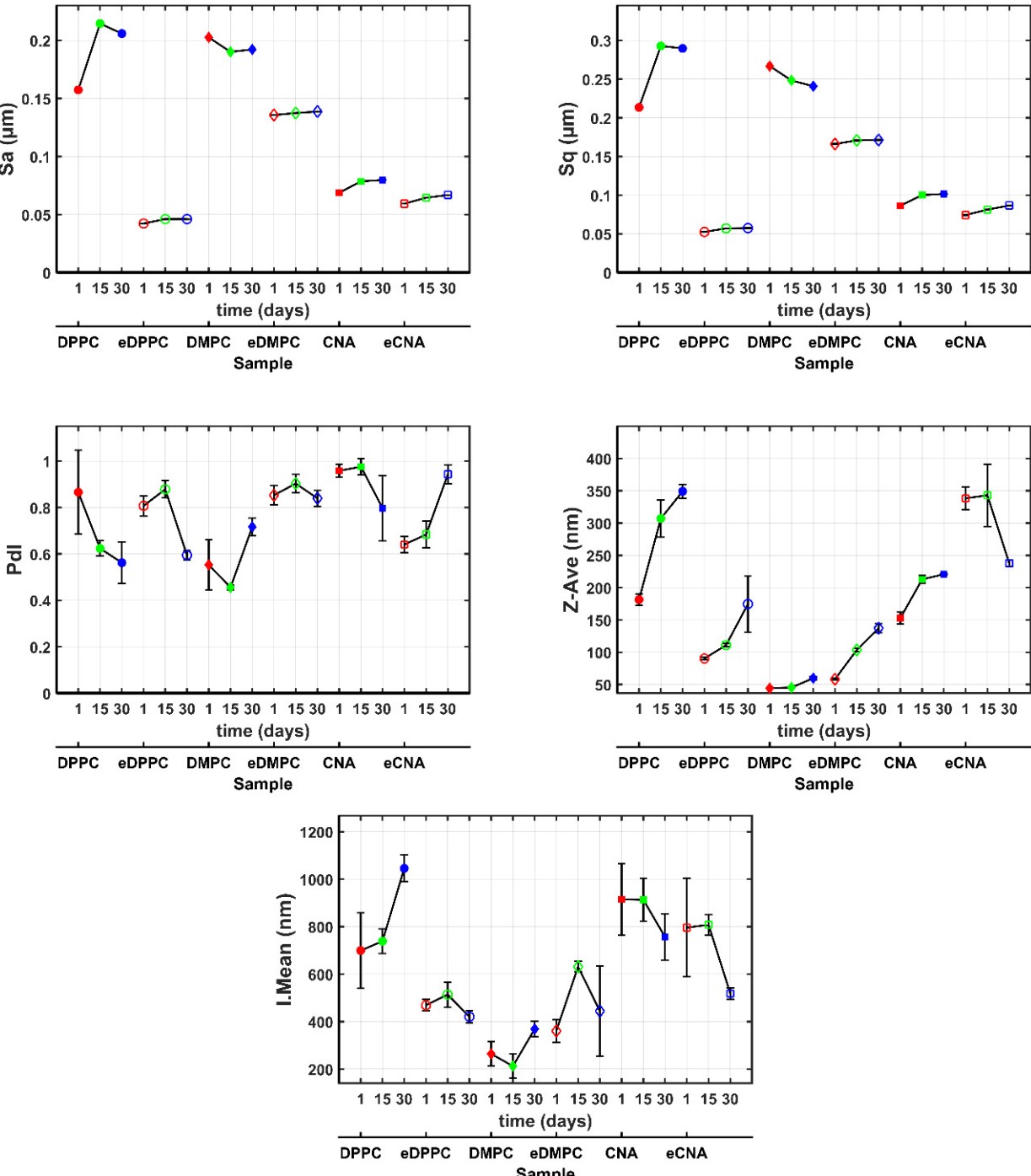

**Figure 9.** Interval plots of the liposome's properties for Sample*Time interaction factor levels; CA—caffeic acid, DPPC, DMPC, CNA—liposomes loaded with CA, and eDPPC, eDMPC, eCNA—empty liposomes.

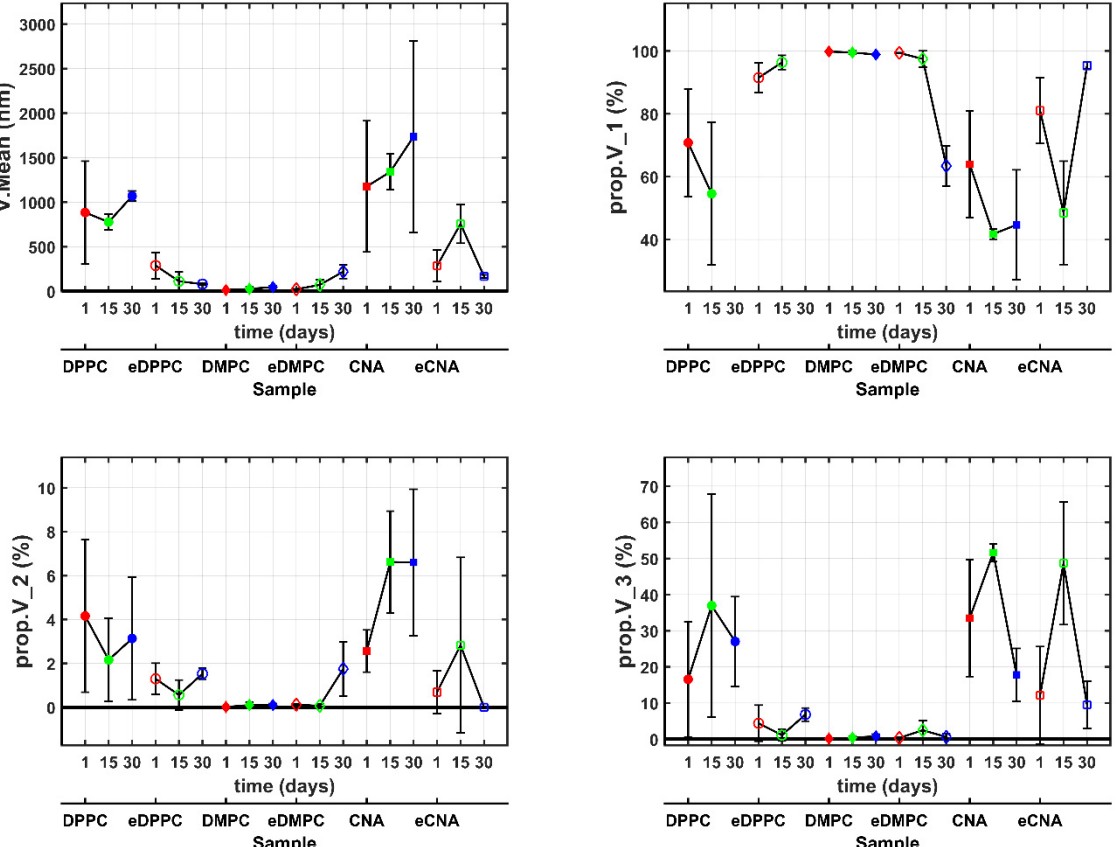

**Figure 10.** Interval plots of the liposome's properties derived from DLS analysis's volume-weighted distribution. Results are displayed for the Sample*Time interaction factor levels. CA—caffeic acid, DPPC, DMPC, CNA—liposomes loaded with CA, and eDPPC, eDMPC, eCNA—empty liposomes.

### 3.6.2. Multivariate Statistical Analysis

As was stated previously, a multivariate statistical sequence was used. It consisted of: PCA, MANOVA ($p = 0.05$) and AHC statistical methods. The aim of the multivariate sequence was to establish which liposome samples had the highest encapsulation effect and high thermodynamic particle size stability. In order to obtain suitable results, it was necessary to generate the proper number, and thus the composition, of the liposome clusters. After the liposome sample clustering, one can correctly interpret the PCA results.

Another important aspect of the multivariate statistical analysis is that only the MANOVA has a statistical significance level ($p = 0.05$) that give a statistical confidence of the results and thus valid conclusions.

The multivariate statistical sequence begins with the PCA method, which generates the principal coordinates (PC1–PC7, Table 9) of studied parameters for all the liposomes (i.e., the Sample*Time levels). The principal coordinates PC1 to PC4 cover 97.436% of cumulative explained variance. However, based on supra-unitary eigenvalue selection criterion, the scree plot (data not shown) prescribes only the first two principal components to be enough to further data interpretation, as the cumulative explained variance is too low (i.e., 79%). In consequence, in order to gain a high level of cumulative explained variance, the retained principal component number was extended to four. However, to simplify the visual PCA biplot interpretation, only PC1 to PC3 were used to draw the ordination conclusions.

The PCA 2D and 3D biplots (Figure 11a,b and Figure 12a,b) display the variable vectors, starting from principal coordinates system origin, and are pointing out the direction that the corresponding variable has the highest abundance or concentration values. The opposite direction, thus, is pointing out the lowest abundance or concentration values.

In this way, PCA is an ordination multivariate statistical method that generates relative comparisons between studied samples based on their parameter values. Furthermore, the variable vectors between the polar angle have small values, meaning that those variables are strongly correlated. If variable vectors are pointing out to some liposome samples, this means that the samples display high level abundance or concentration of the variables—and vice versa.

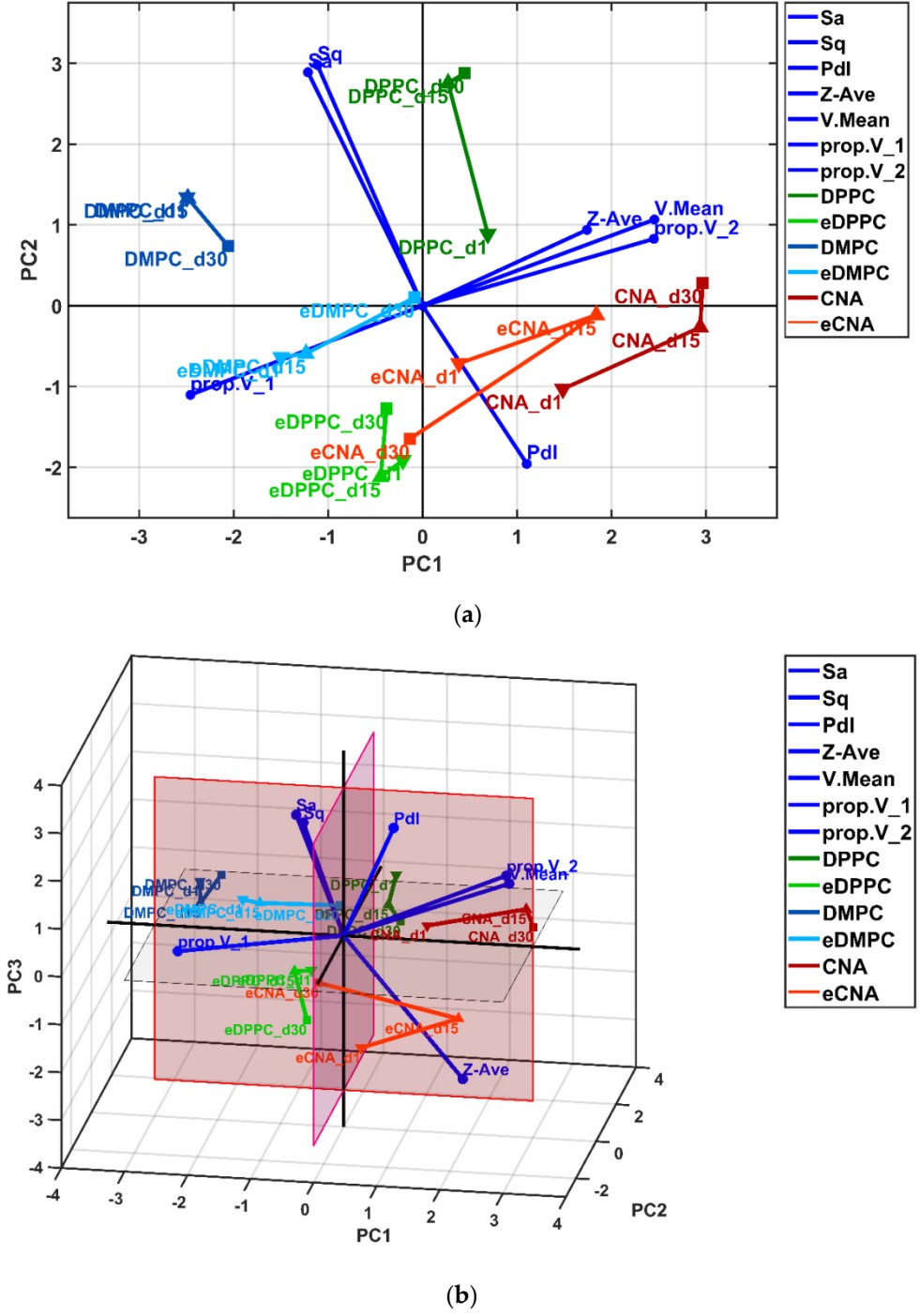

**Figure 11.** Biplots of PCA results: (**a**) 2D representation for PC1 and PC2 principal components; (**b**) 3D representation for PC1, PC2 and PC3 principal components. CA—caffeic acid, DPPC, DMPC, CNA—liposomes loaded with CA, and eDPPC, eDMPC, eCNA—empty liposomes.

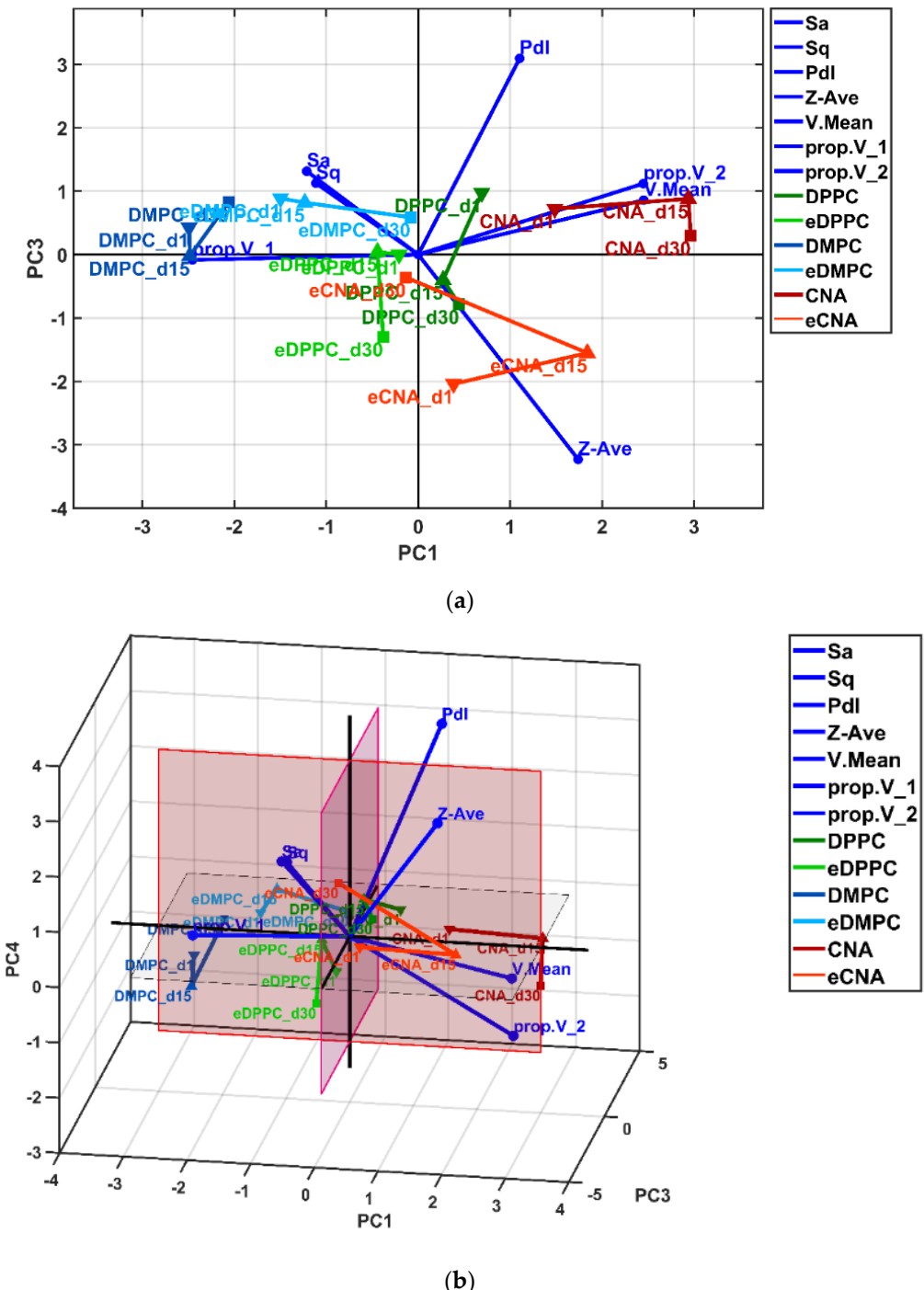

**Figure 12.** Biplots of PCA results: (**a**) 2D representation for PC1 and PC3 principal components; (**b**) 3D representation for PC1, PC3 and PC4 principal components. CA—caffeic acid, DPPC, DMPC, CNA—liposomes loaded with CA, and eDPPC, eDMPC, eCNA—empty liposomes.

The PC1–PC2 corresponding principal coordinates were considered as numerical inputs for the MANOVA ($p = 0.05$) test method. The emerging results of this method are: a matrix of statistical significances of the pairwise sample comparisons (Bonferroni corrected values, $p = 0.05$) and the canonical coordinates (Canon1–Canon7) of the variables for all the liposome samples.

Usually, the canonical coordinates are calculated to increase the variance-covariance level and thus to generate highest possible Euclidean distance between the samples. In consequence, in order to obtain the liposomes clusters, the AHC clustering method was

used on the Canon1 to Canon4 coordinate values. The AHC results are presented as heat maps with a dendrogram (Figure 13a) and a simple dendrogram with the cut-off similarity distance value (Figure 13b) used to obtain the proper number of clusters, based on MANOVA's *p*-values matrix (Table 10).

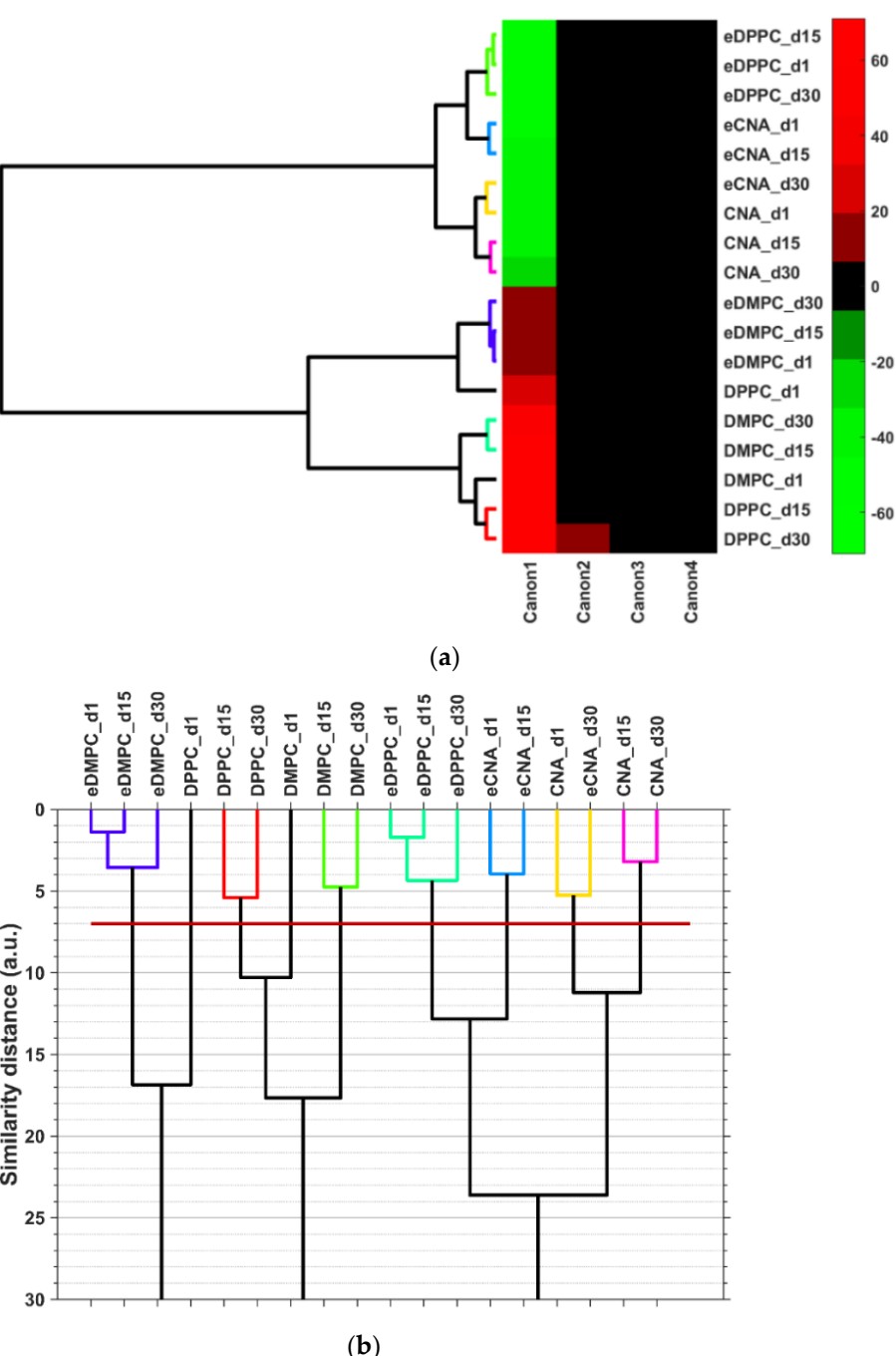

**Figure 13.** Cluster analysis of MANOVA (*p* = 0.05) results: (**a**) Heatmap representation for Canon1 ÷ Canon4 canonical coordinates; (**b**) Dendrogram representation for Canon1 ÷ Canon4 canonical coordinates with the cut-off red line that generates the clusters. CA—caffeic acid, DPPC, DMPC, CNA—liposomes loaded with CA, eDPPC, eDMPC, eCNA—empty liposomes, d1—day 1, d15—day 15, d30—day 30.

**Table 10.** Statistical significance *p*-values generated by MANOVA (*p* = 0.05) method in post-hoc pairwise multiple comparisons.

| *p*-Values | DPPC _d1 | DPPC _d15 | DPPC _d30 | eDPPC _d1 | eDPPC _d15 | eDPPC _d130 | DMPC _d1 | DMPC _d15 | DMPC _d30 | eDMPC _d1 | eDMPC _d15 | eDMPC_d30 | CNA_d1 | CNA_d15 | CNA_d30 | eCNA_d1 | eCNA_d15 | eCNA_d30 |
|---|---|---|---|---|---|---|---|---|---|---|---|---|---|---|---|---|---|---|
| DPPC_d1 | | 0.000 | 0.000 | 0.000 | 0.000 | 0.000 | 0.000 | 0.000 | 0.000 | 0.000 | 0.000 | 0.000 | 0.000 | 0.000 | 0.000 | 0.000 | 0.000 | 0.000 |
| DPPC_d15 | 0.000 | | 0.0847 | 0.000 | 0.000 | 0.000 | 0.000 | 0.000 | 0.000 | 0.000 | 0.000 | 0.000 | 0.000 | 0.000 | 0.000 | 0.000 | 0.000 | 0.000 |
| DPPC_d30 | 0.000 | 0.0847 | | 0.000 | 0.000 | 0.000 | 0.000 | 0.000 | 0.000 | 0.000 | 0.000 | 0.000 | 0.000 | 0.000 | 0.000 | 0.000 | 0.000 | 0.000 |
| eDPPC_d1 | 0.000 | 0.000 | 0.000 | | 1.000 | 0.000 | 0.000 | 0.000 | 0.000 | 0.000 | 0.000 | 0.000 | 0.000 | 0.000 | 0.000 | 0.000 | 0.000 | 0.000 |
| eDPPC_d15 | 0.000 | 0.000 | 0.000 | 1.000 | | 0.000 | 0.000 | 0.000 | 0.000 | 0.000 | 0.000 | 0.000 | 0.000 | 0.000 | 0.000 | 0.000 | 0.000 | 0.000 |
| eDPPC_d30 | 0.000 | 0.000 | 0.000 | 0.000 | 0.000 | | 0.000 | 0.000 | 0.000 | 0.000 | 0.000 | 0.000 | 0.000 | 0.000 | 0.000 | 0.000 | 0.000 | 0.000 |
| DMPC_d1 | 0.000 | 0.000 | 0.000 | 0.000 | 0.000 | 0.000 | | 0.1202 | 0.000 | 0.000 | 0.000 | 0.000 | 0.000 | 0.000 | 0.000 | 0.000 | 0.000 | 0.000 |
| DMPC_d15 | 0.000 | 0.000 | 0.000 | 0.000 | 0.000 | 0.000 | 0.1202 | | 0.0523 | 0.000 | 0.000 | 0.000 | 0.000 | 0.000 | 0.000 | 0.000 | 0.000 | 0.000 |
| DMPC_d30 | 0.000 | 0.000 | 0.000 | 0.000 | 0.000 | 0.000 | 0.000 | 0.0523 | | 0.000 | 0.000 | 0.000 | 0.000 | 0.000 | 0.000 | 0.000 | 0.000 | 0.000 |
| eDMPC_d1 | 0.000 | 0.000 | 0.000 | 0.000 | 0.000 | 0.000 | 0.000 | 0.000 | 0.000 | | 0.9579 | 0.000 | 0.000 | 0.000 | 0.000 | 0.000 | 0.000 | 0.000 |
| eDMPC_d15 | 0.000 | 0.000 | 0.000 | 0.000 | 0.000 | 0.000 | 0.000 | 0.000 | 0.000 | 0.9579 | | 0.000 | 0.000 | 0.000 | 0.000 | 0.000 | 0.000 | 0.000 |
| eDMPC_d30 | 0.000 | 0.000 | 0.000 | 0.000 | 0.000 | 0.000 | 0.000 | 0.000 | 0.000 | 0.000 | 0.000 | | 0.000 | 0.000 | 0.000 | 0.000 | 0.000 | 0.000 |
| CNA_d1 | 0.000 | 0.000 | 0.000 | 0.000 | 0.000 | 0.000 | 0.000 | 0.000 | 0.000 | 0.000 | 0.000 | 0.000 | | 0.8200 | 0.0559 | 0.000 | 0.000 | 0.7627 |
| CNA_d15 | 0.000 | 0.000 | 0.000 | 0.000 | 0.000 | 0.000 | 0.000 | 0.000 | 0.000 | 0.000 | 0.000 | 0.000 | 0.8200 | | 0.5454 | 0.000 | 0.000 | 0.023 |
| CNA_d30 | 0.000 | 0.000 | 0.000 | 0.000 | 0.000 | 0.000 | 0.000 | 0.000 | 0.000 | 0.000 | 0.000 | 0.000 | 0.0559 | 0.5454 | | 0.000 | 0.000 | 0.000 |
| eCNA_d1 | 0.000 | 0.000 | 0.000 | 0.000 | 0.000 | 0.000 | 0.000 | 0.000 | 0.000 | 0.000 | 0.000 | 0.000 | 0.000 | 0.000 | 0.000 | | 0.7901 | 0.000 |
| eCNA_d15 | 0.000 | 0.000 | 0.000 | 0.000 | 0.000 | 0.000 | 0.000 | 0.000 | 0.000 | 0.000 | 0.000 | 0.000 | 0.000 | 0.000 | 0.000 | 0.7901 | | 0.000 |
| eCNA_d30 | 0.000 | 0.000 | 0.000 | 0.000 | 0.000 | 0.000 | 0.000 | 0.000 | 0.000 | 0.000 | 0.000 | 0.000 | 0.7627 | 0.000 | 0.000 | 0.000 | 0.000 | |

CA—caffeic acid, DPPC, DMPC, CNA—liposomes loaded with CA, and eDPPC, eDMPC, eCNA—empty liposomes, d1—day 1, d15—day 15, d30—day 30.

The heat map representation emphasizes the canonical coordinate property to increase the between-sample Euclidean distances. This property is validated by higher Canon1 and Canon2 coordinate values than the Canon3 and Canon4 ones. All these results conduct to the clustering information presented in Table 11 and Figure 13a,b.

**Table 11.** AHC results on Canon1÷Canon4, MANOVA's canonical coordinates.

| Clusters | Sample*Time Factor Levels | MANOVA's Canon1 ÷ Canon4, AHC Liposomes Grouping |
|:---:|:---:|:---:|
| C1 | DPPC_d1 | DPPC_d1 |
| C2 | DPPC_d15 DPPC_d30 | DPPC_d15; DPPC_d30 |
| C3 | eDPPC_d1 eDPPC_d15 | eDPPC_d1; eDPPC_d15 |
| C4 | eDPPC_d30 | eDPPC_d30 |
| C5 | DMPC_d1 DMPC_d15 | DMPC_d1; DMPC_d15 |
| C6 | DMPC_d30 | DMPC_d30 |
| C7 | eDMPC_d1 eDMPC_d15 | eDMPC_d1; eDMPC_d15 |
| C8 | eDMPC_d30 | eDMPC_d30 |
| C9 | CNA_d1 CNA_d15 CNA_d30 | CNA_d1; CNA_d15; eCNA_d30 CNA_d15; CNA_d1; CNA_d30 CNA_d30; CNA_d15 |
| C10 | eCNA_d1 eCNA_d15 | eCNA_d1; eCNA_d15 |
| C9 | eCNA_d30 | eCNA_d30; CNA_d1 |

CA—caffeic acid, DPPC, DMPC, CNA—liposomes loaded with CA, eDPPC, eDMPC, eCNA—empty liposomes, d1—day 1, d15—day 15, and d30—day 30.

For the 2D and 3D biplot with clusters (Figure 14 a,b) the liposomes at nano-level (1) with high roughness and CA encapsulated volume are: DMPC_d1, DMPC_d15, DMPC_d30, eDMPC_d1, eDMPC_d15 and eDMPC_d30; at nano-level (2) with intermediate roughness level are: DPPC_d1, DPPC_d15, DPPC_d30, eDPPC_d1, eDPPC_d15, eDPPC_d30, DMPC_d1, DMPC_d15, DMPC_d30, eDMPC_d1, eDMPC_d15, eDMPC_d30, CNA_d1, CNA_d15, CNA_d30, eCNA_d1, eCNA_d15 and eCNA_d30; finally, at nano-level (2) with low roughness level are: CNA_d1, CNA_d15, CNA_d30, eCNA_d1 and eCNA_d15.

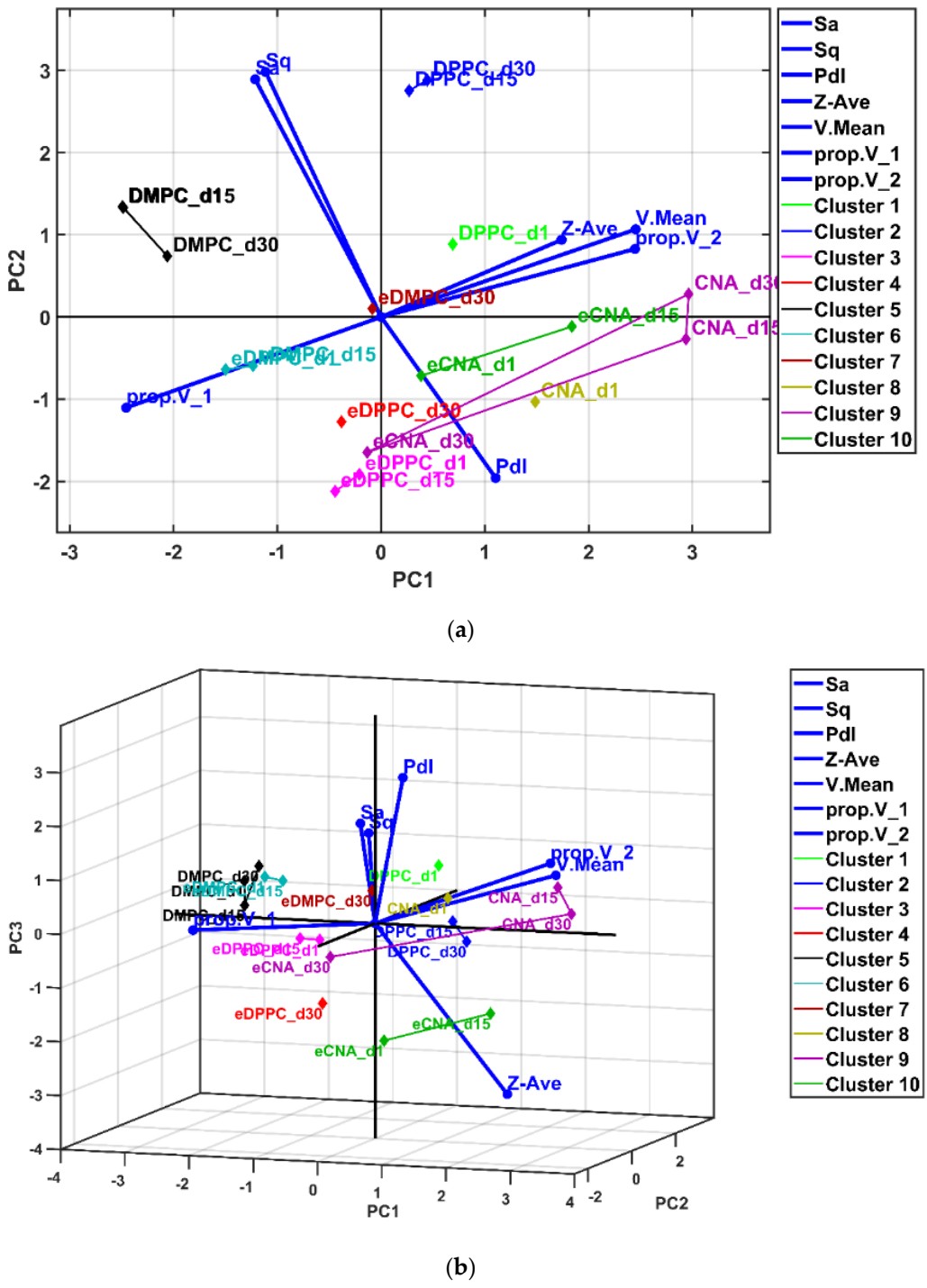

**Figure 14.** Biplots of PCA results with clusters in original PCA coordinates: (**a**) 2D representation for PC1 and PC2 principal components; (**b**) 3D representation for PC1, PC2 and PC3 principal components. CA—caffeic acid, DPPC, DMPC, CAN—liposomes loaded with CA, eDPPC, eDMPC, eCNA—empty liposomes, d1—day 1, d15—day 15, and d30—day 30.

## 4. Discussions

The characterization of liposomes loaded with CA was performed at the beginning by analysing the macroscopic appearance of the liposomal suspension. It has been observed that a suspension containing liposomes has a milky, pale yellow appearance. Then the liposomes loaded with CA were characterized by their size, zeta potential, EE and their morphology.

Analysing the data obtained on EE, a high entrapment of CA in liposomes was observed; this may be due to the low solubility of CA in water at room temperature. Following the evaluation of EE, no significant influences were observed between the three liposome formulas, even if various structures of phospholipids or sodium cholate were used, which is consistent with other studies [42]. A high CA entrapment (70 ± 4%) and a different potential (−55 ± 4 mV) were obtained if reverse phase evaporation technique and only phosphatidylcholine were used for for liposome formation [43]. Pettinato et al. [29] extracted antioxidants from spent coffee grounds followed by entrapping the extract into liposomes using supercritical assisted liposome formation, resulting in good encapsulation with 93% of the loading antioxidant activity.

There are studies which have shown that the higher zeta potential is, the more stable the liposomes are [44]. Taking this into account and analysing the data obtained for the zeta potential, it was observed that the DPPC, DMPC and CNA liposomes are more stable than the corresponding liposomes loaded without CA (eDPPC, eDMPC and eCNA). Comparing three liposomes loaded with CA, results showed that the best stability is obtained by the CNA liposome, followed by DMPC and then DPPC. The modest values of zeta potentials indicate a low stability of the nanoparticles, which transform to bigger nanoparticles during storage.

The eDPPC and DPPC liposomes present the same four-peak shape distribution. There is present a bimodal distribution of nano-level (1) particles (d.nm < 100 nm), followed by very low-volume proportion of nano-level (2) particles and a low-volume proportion of nano-level (3) (Figure 2) particles. The difference between them is the proportions of volume particles of each nano-level. The eDPPC, without encapsulated CA, presents a lower-volume concentration of large particles (nano-level (2)) and flocculated liposomes than the DPPC liposomes with encapsulated CA. This result indicates that CA facilitates the aggregation of DPPC liposomes.

The eDMPC and DMPC liposomes present, a unimodal and bimodal peak shape distribution of nano-level particles (d.nm < 100 nm), and a very low-volume proportion (Figure 3). From these distributions, both eDMPC and DMPC present the same high-volume of nano-level (1) particles and a very low-volume concentration of large particles (nano-level (2)) and flocculated liposomes. This result indicates that DMPC liposomes are insensitive to aggregation in the presence of CA.

The eCNA and CNA liposomes present the same four-peak shape distribution. There is present a bimodal distribution of nano-level (1) particles (d.nm < 100 nm) with high-volume proportion. This is followed by a low-volume proportion of nano-level (2) particles and a medium-high volume proportion of nano-level (3) particles (Figure 4). The CNA presents low volume concentration of large particles (nano-level (2)) and medium-high volume concentration of flocculated liposomes. Furthermore, the bimodal peak of CNA liposome has its second peak (with greater d.nm) as the highest, being higher than the eCNA liposome. This result prescribes that CA facilitates the aggregation and flocculation of CNA liposomes.

From the DLS panel results, the following were retained for further statistical analysis: Z-Ave (d.nm), which represents the intensity-weighted mean intensity particle diameter from a gaussian distribution that approximates the measured diameter range of each liposome sample; PdI, the polydispersity index (the lowest the higher uniformity of the particle sizes); V.mean (d.nm) and I.mean (d.nm), the mean values of particle diameters (nm) derived from the volume-weighted and intensity-weighted distributions of particle sizes, respectively; and prop.V_1, prop.V_2 and prop.V_3, the proportions (%) of the particle volume concentrations, each calculated for particle nano-levels (1), (2) and (3).

At analysis through AFM, for the sample DPPC and eDPPC, on the first and the second day the obtained results show similar values for Sa, Sq, Sp, Sv, Sy, and an increase of the viscosity and a higher maximum value being observed only on the second day. Comparing the 15th day with the 30th, higher roughness values were observed for the 15th day. Sp and Sy values increased further on the 30th day.

For eDPPC, the roughness values (Sa and Sq) remained the same on the first and second day, and on the 15th and 30th very small, almost insignificant changes were observed. Given that the force applied to the tip on the surface and the distance between the tip and the surface have the same values, some changes may occur due to the nature of the sample analysed in combination with the tip wear [45].

Analyses on the roughness of the DMPC sample shows that the values decreased over time. Increasing values were detected in case of Sy regarding the first three measured periods, whereas in the last day (30th day) extre me decrease was observed. Sp shows similar tendency as Sy, while for the Sv different tendencies exist during periods but the last one (30th) exhibited the same decrease as in the previously mentioned parameters.

The analyses obtained for the eDMPC sample show that the roughness value increased over time. However, the values are almost similar. In Sp, Sv and Sy case, the values from the 15th day are the highest. Sv presents a general decrease, except for the 15th day. Sy presents a general increase of values in time. Sp shows increasing tendency for the first three measurements, while in the last measurement a decrease is observed.

When determining the efficiency of entrapment, it was shown that three liposomes obtained with different phospholipids entrapped amounts close to CA, the highest amount being at the CNA liposome.

In vitro release of CA in Franz diffusion cells showed a close release as a percentage for each liposome, but the highest percentage of CA release was for the DPPC liposome (89.653%) (Table 6). It can be seen that after 4 h of diffusion, the percentages of CA release incorporated in liposomes reached 45%. Over the next 3 h, the percentages increased up to 81%. After 12 h, a very small increase (max 3%) was observed. The rest of the liposomal CA could be permanently trapped inside the liposomes [43].

At the CA free diffusion, it was observed that after 4 h, the diffusion percentage was 92.534%. By comparing the diffusion, it can be concluded that liposomes ensure the maintenance of CA concentrations for a longer period of time (more than 7 h).

In the univariate statistical analysis, the Sample factor results averages all the values of the Time factor sample for each sample (Table 7). The possible drawn conclusion, then, will describe the nano-levels and stability over time of the Sample factor levels. All three liposomes with CA encapsulated have higher stability (i.e., higher Sa and Sq values) than the corresponding liposomes without CA encapsulation. High roughness is present for the DPPC and DMPC liposomes, higher than CNA.

The polydispersity index for DPPC and DMPC type liposomes increases for CA encapsulated samples compared with corresponding samples without CA encapsulation (Table 7 and Figure 7). However, this increase is not statistically significant, indicating that from a thermodynamic point of view the samples are stable. The CNA liposome changes thermodynamic stability when the CA is encapsulated, when the sample becomes more stable.

The volume-weighted particle size distributions for DPPC and eDPPC, CNA and eCNA and DMPC-type liposomes present high PdI values (over 0.60) that validate the multimodal particle size distribution. For this kind of multimodal behaviour of the liposomes the Z-Ave and PdI parameters are the most suitable to make comparisons between sample particle sizes.

For qualitative comparisons, a statistically significant increase of liposome diameter is displayed after CA encapsulation by DPPC and CNA type liposomes, above five times and three times, respectively. Furthermore, despite the fact DMPC liposome diameters increased 3.6 times after CA encapsulation, this increase is statistically not significant.

However, these results prescribe high levels of CA encapsulation, but come with an issue that the liposome nano-level changes up one level. From the medical cell cancer treatment point of view, these results are more than beneficial.

The possible drawn conclusion, then, will prescribe the nano-levels and their stability of the Sample factor levels at each studied time. The univariate analysis of these results would take too many textual resources and should be approximatively redundant with the

Sample factor analysis results (from Table 8 and Figures 7 and 8). The only new information is the time behaviour of each sample, but with the same overall conclusions. Furthermore, these results are needed for cross-validation with the next multivariate statistical analysis. The following parameters values were used in the multivariate analysis: Sa, Sq, PdI, Z-Ave, prop.V_1, prop.V_2, prop.V_3 and V.mean.

From the PC1–PC2 and PC1–PC3 biplots (Figures 11a and 12a), variable correlations that generate variable groups can be noted, as follows: Sa with Sq; V.mean with prop.V_2 and singletons of Z-Ave, PdI and prop.V_1. As expected, the Z-Ave, prop.V_1, prop.V_2 and PdI variable vectors are almost spatially (i.e., 3D) opposite as directions (Figure 11a,b and Figure 12a,b). This result denotes the PCA biplot as a "shell-type" variable distribution and generates mostly good group sample separation.

The generated 2D and 3D PCA biplots also display the Sample factor level trajectories over time (day 1, day 15 and day 30). The longer the trajectory is in time, the less multivariate stable the sample is. The most stable samples are for both the DMPC- and eDPPC-type liposomes. Intermediate stability is seen for DPPC- and can-type liposomes; finally, the leas-stable sample is the eCNA-type liposome.

From a medical point of view, the direct interest should be the CA-encapsulated liposomes at day 1 timestamp. DPPC_d1 and DMPC_d1 can be considered nanoparticle liposomes with high stability, that present 75% and 99% cumulative particle volumes of 30–500 nm diameters. The CNA_d1 liposome provided only 66% particles cumulative volume at 30–500 nm diameters with a "soft" carrier layer, thus being unstable over time. However, if the liposomes are embedded in a jellified lattice, then all the CA-encapsulated liposomes can be used as nanoparticle treatments.

## 5. Conclusions

From the starting concept that liposomes are an important part of medical and pharmaceutical research, being considered to be among the most effective carriers for the introduction of various drugs into target cells, in this study we aimed to achieve three types of liposomes prepared by hydration of the lipid film. In this sense, we used combinations of phospholipids in which cholesterol was used as a stabilizing agent. The same amount of CA was trapped in each type of formula, resulting in nanometric particle structures with variable dimensions, being a mixture of medium and giant liposomes. Based on the results obtained by AFM and DLS, we observed the polydispersity of liposomes and we can say that liposomes based on DPPC and DMPC can be considered to have high stability. In addition, these formulas have nanometric dimensions, and about 75–99% had dimensions between 40–500 nm. The larger size of the other liposome formula confirmed that the type of phospholipids used for the preparation significantly influenced their size and electrical charge. For the characterization of liposomes, we evaluated the entrapment capacity and the release, being up to 70% CA and ensuring a slow release. Therefore, liposomes offer great potential in CA entrapment.

Taking into account the activity of CA to prevent premature aging of the skin, we aim to continue research by incorporating these liposomal particulate structures in preparations with dermal application. We believe that in this way we will be able to obtain ointment formulas that can ensure an optimal concentration of CA in dermal tissue.

**Author Contributions:** Conceptualization, methodology and writing of the research were done by I.L.D., L.G.V., A.C.T., L.F., P.S., G.A.G., G.E.D., O.A.M., A.S.B. did experimentation and data analysis. T.J., M.E.M., A.P. and E.M. did supervision, editing and review. All authors have read and agreed to the published version of the manuscript.

**Funding:** This research received no external funding.

**Institutional Review Board Statement:** Not applicable.

**Informed Consent Statement:** Not applicable.

**Data Availability Statement:** Data available in a publicly accessible repository.

**Acknowledgments:** The author (Dejeu Ioana Lavinia) thanks to the University of Oradea, Doctoral School of Biomedical Science, for providing assistance.

**Conflicts of Interest:** The authors declared no potential conflict of interest concerning the research, authorship, and/or publication of this article.

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
