# Peer review of "Liposomes with Caffeic Acid: Morphological and Structural Characterisation, Their Properties and Stability in Time"

_processes, doi:10.3390/pr9060912_

Round 1
Reviewer 1 Report
Authors provided a paper entitled “Liposomes with caffeic acid: morfological and structural characterisation, their properties and 2 stability in time” for the publication on Processes, MDPI.
This paper has a good scientific soundness and deserves to be published after major revisions.
The use of English is good.
I suggest adding an abbreviation list of acronyms, according to Processes guidelines.
Here is the list of my issues:
Line 22. “ In this study we obtained six liposome formulas”. I would say “we prepared”
Line 25. “acid entrapment is good (76%).” I would just say that encapsulation efficiency is 76 % or up to 76 %.
Line 27. “ liposomes have medium and giant dimensions”. I suggest indicating precise mean dimensions and standard deviations.
In my opinion, Figure 1 is not necessary since the molecule structure is well known.
Line 42 “Can be isolated from thyme, oregano, sage, berries, apple, coffee, potato”. this sentence has a missing subject.
Line 48. check and remove a double space here.
Line 59. “Liposomes are composed of cholesterol and phospholipids”. I would not just say this sentence as definition. Cholesterol is often introduced into liposomes, but it is not the rule. sometimes, different kind of additives are included into the double lipidic layers.
Lines 68-69. “thin film hydration method and solvent removal method”. could you please add references for this?
Line 71. “from 50 to 450 nm.”. please add a reference here. also in this case, it is not the general rule. some pharmaceutical companies say that to maintain sterile conditions, mean dimensions lower than 200 nm are needed.
Line 84. “In this paper we incorporated-…” the aims of this paper are described in just two lines. I suggest improving this part, developing it.
Line 106. please add the country of the manufacturer of the rotavapor.
Line 119. “with a UV-VIS spectrophotometer”. I suggest substituting “with” with “using”. and “a” with “an”.
“Paragraph 2.2 Determination of entrapment efficiency (EE%).” I suggest adding the equation for the calculation of encapsulation efficiency, starting from the absorbance value. I have noticed that this equation has been indicated at line 231. However, it could be better to move this equation to material and methods section.
Line 143. “1-150 nm,”. I would not start the first range as 1 nm. I would at least propose 30-40 nm, not less, in my opinion.
Line 162. a double space has been detected here.
Line 216 “in which caffeic acid was entrapped”. I would say “…loaded with caffeic acid”.
Table 2 and Table 3 could be unified in a proper manner, providing simultaneously mean dimensions and zeta potential.
Table 2. I think that 2 decimal digits are sufficient for encapsulation efficiency.
Line 283. “was used the calibration curve of the caffeic acid” could be better as “the calibration curve of caffeic acid was used…”
Figure 7. I suggest comparing these 3 release profiles on the same diagrams. It could be better to read the information proposed and make comparisons. Be careful, the denomination of Figure “c” is not reported as “(c)” but as “©”
Table 7. By commenting this table, could you define the initial entrapped concentration of caffeic acid?
Table 8. I think that no decimal digits are necessary to define mean dimensions and standard deviations since they are already defined in terms of nanometers. I suggest removing decimal digits, also because this table needs more space to be compacted.
Very good statistical analysis.
Line 446. “premise”. maybe better “introduction” or maybe “starting concept”, something similar.
Line 456. “1 to 500 nm”. Again, I would start from 40-50 nm.
Line 460. “trapped over 70%”. I would say “up to …”
Line 462. “This section may be divided by subheadings. It should provide a concise and precise description of the experimental results, their interpretation, as well as the experimental conclusions that can be drawn”
maybe this paragraph belongs to the template for this journal. Authors maybe can eliminate it.
Some lines about future perspectives could be added.
Let me please suggest the reading of this paper “Pettinato, M., Trucillo, P., Campardelli, R., Perego, P., & Reverchon, E. (2020). Bioactives extraction from spent coffee grounds and liposome encapsulation by a combination of green technologies. Chemical Engineering and Processing-Process Intensification, 151, 107911.”
Another reading could be useful: Arya, S. S., Venkatram, R., More, P. R., & Vijayan, P. (2021). The wastes of coffee bean processing for utilization in food: a review. Journal of Food Science and Technology, 1-16.
Author Response
Thank you for your suggestiona.
In this paper, we wrote in red all your suggestions to make them more visible.

Reviewer 2 Report
Article entitled "Liposomes with caffeic acid: morphological and structural characterization, their properties and stability in time" presents production and assessment of liposomes containing caffeic acid. The authors prepared and analyzed six formulations in total, in which half were placebo, and half included API.
1. The study's goal in the introduction section is formulated wrong as CA was added in the same concentration in every sample. The goal of the study should be clearly presented.
2. Within the introduction, section authors should provide information from literature about other research related to the topic of CA liposome preparations (there are a few of them!) These should also build a base on further discussion, which is completely lacking in the manuscript.
3. Standard deviations for Zeta potential and surface assessment should be provided to make comparison possible.
4. Did the authors try to measure EE using the indirect method by ultracentrifuge sample and measure the amount of drug in supernatant? I believe that chosen method is overcomplicated and not fully sure about its validity - maybe it is caused by flaws in the description.
5. Abbreviations should be explained in tables and figures description.
The quality of plots must be improved - I was not able to read captions.
6. Extensive English editing is required to make the text understandable and easy to follow for readers.
7. For API release study CA (non-liposomal form) as a control sample should be introduced.
Author Response
Thank you for your suggestiona.
In this paper, we wrote in red all your suggestions to make them more visible

Round 2
Reviewer 1 Report
Authors provided a revised version of the paper, that now is in a much improved version.
I only have some minor additional issues:
Line 27. 40±0.55 – 500±1.45 nm. the font size of standard deviation is larger than mean value. check it.
This was my previous comment. “Lines 68-69. “thin film hydration method and solvent removal method”. could you please add references for this?” Did authors add reference to this lines?
Line 82. the release rate is optimizes. check the English here.
From line 84 to 88. This should be in materials and methods section.
At the end of the introduction section, a paragraph on the aims of this paper should be.
Line 130. Equation should be numbered.
Thank you
Author Response
In this paper, we wrote in bleu all your suggestions to make them more visible.
Thanks for the suggestions.
Line 27. 40±0.55 – 500±1.45 nm. the font size of standard deviation is larger than mean value. check it. – We corrected
This was my previous comment. “Lines 68-69. “thin film hydration method and solvent removal method”. could you please add references for this?” Did authors add reference to this lines? – We added [19-21]
- Butu, A., S. Rodino, D. Golea, M. Butu, M. Butnariu, C. Negoescu, and C.E. Dinu-Pirvu, LIPOSOMAL NANODELIVERY SYSTEM FOR PROTEASOME INHIBITOR ANTICANCER DRUG BORTEZOMIB. Farmacia, 2015. 63(2): p. 224-229.
- Karpuz, M., E. Atlihan-Gundogdu, E.S. Demir, and Z. Senyigit, Radiolabeled Tedizolid Phosphate Liposomes for Topical Application: Design, Characterization, and Evaluation of Cellular Binding Capacity. Aaps Pharmscitech, 2021. 22(2).
- Wang, W., G.-f. Shu, K.-j. Lu, X.-l. Xu, M.-c. Sun, J. Qi, Q.-l. Huang, W.-q. Tan, and Y.-z. Du, Flexible liposomal gel dual-loaded with all-trans retinoic acid and betamethasone for enhanced therapeutic efficiency of psoriasis. Journal of Nanobiotechnology, 2020. 18(1): p. 80.
Line 82. the release rate is optimizes. check the English here. – We took out
From line 84 to 88. This should be in materials and methods section. – We corrected
At the end of the introduction section, a paragraph on the aims of this paper should be. – We added
Line 130. Equation should be numbered. – We numbered.
Reviewer 2 Report
Dear Authors,
I appreciate changes that were made according to suggestions provided to the prior version of the manuscript. For now, it is easier to understand the general idea and interpret provided results. However, there are still minor changes required.
- Presenting the general goal of the research, I recommend modifying sentences in lines: 79-84. First of all, the authors made this liposome formulation to protect CA from oxidative decomposition. Your general idea is targeting the cosmetic industry and topical application, as was pointed in the final sentence. The second comment is related to optimization therm. It is difficult to say that formulations were optimized in terms of API release as there were just three preliminary samples and no one we can call optimized. Moreover, dissolution from all of them was quite identical. Correcting this, please take care about grammar construction as within the revised version, authors mentioned that extensive English correction was made and still quite annoying errors occur.
- Figures 2,3,4,5 are still not possible to read - I recommend higher resolution pictures.
- Line 406-408 is not relevant to research and presented content as bioavailability was not tested at all.
- Within discussion as within introduction, there are still annoying mistakes occurs like line 418 (should be CA not AC to be consistent through manuscript)
Author Response
In this paper, we wrote in bleu all your suggestions to make them more visible.
Thanks for the suggestions.
Presenting the general goal of the research, I recommend modifying sentences in lines: 79-84. First of all, the authors made this liposome formulation to protect CA from oxidative decomposition. Your general idea is targeting the cosmetic industry and topical application, as was pointed in the final sentence. The second comment is related to optimization therm. It is difficult to say that formulations were optimized in terms of API release as there were just three preliminary samples and no one we can call optimized. Moreover, dissolution from all of them was quite identical. Correcting this, please take care about grammar construction as within the revised version, authors mentioned that extensive English correction was made and still quite annoying errors occur.
We modified and corrected where was necessarily.
- Figures 2,3,4,5 are still not possible to read - I recommend higher resolution pictures. – We modified the resolution of the figures 2,3,4,5.
- Line 406-408 is not relevant to research and presented content as bioavailability was not tested at all. – We took out.
- Within discussion as within introduction, there are still annoying mistakes occurs like line 418 (should be CA not AC to be consistent through manuscript) – We modified.